# Single-cell profiling of lncRNAs in human germ cells and molecular analysis reveals transcriptional regulation of LNC1845 on LHX8

Nan Wang[1†], Jing He[1†], Xiaoyu Feng[1†], Shengyou Liao[2], Yi Zhao[2], Fuchou Tang[3], Kehkooi Kee[1,4*]

[1]Center for Stem Cell Biology and Regenerative Medicine, Department of Basic Medical Sciences, School of Medicine, Tsinghua University, Beijing, China; [2]Key Laboratory of Intelligent Information Processing, Advanced Computer Research Center, Institute of Computing Technology, Chinese Academy of Sciences, Beijing, China; [3]Biodynamic Optical Imaging Center & Department of Obstetrics and Gynecology, College of Life Sciences, Third Hospital, Peking University, Beijing, China; [4]Tsinghua-Peking Center for Life Sciences, School of Life Sciences, Tsinghua University, Beijing, China

**\*For correspondence:**
kkee@tsinghua.edu.cn

[†]These authors contributed equally to this work

**Competing interest:** The authors declare that no competing interests exist.

**Abstract** Non-coding RNAs exert diverse functions in many cell types. In addition to transcription factors from coding genes, non-coding RNAs may also play essential roles in shaping and directing the fate of germ cells. The presence of many long non-coding RNAs (lncRNAs) which are specifically expressed in the germ cells during human gonadal development were reported and one divergent lncRNA, *LNC1845*, was functionally characterized. Comprehensive bioinformatic analysis of these lncRNAs indicates that divergent lncRNAs occupied the majority of female and male germ cells. Integrating lncRNA expression into the bioinformatic analysis also enhances the cell-type classification of female germ cells. Functional dissection using in vitro differentiation of human pluripotent stem cells to germ cells revealed the regulatory role of *LNC1845* on a transcription factor essential for ovarian follicle development, *LHX8*, by modulating the levels of histone modifications, H3K4me3 and H3K27Ac. Hence, bioinformatical analysis and experimental verification provide a comprehensive analysis of lncRNAs in developing germ cells and elucidate how an lncRNA function as a *cis* regulator during human germ cell development.

## Editor's evaluation

This manuscript provides a comprehensive analysis of expression patterns and genomic features of long non-coding RNAs (lncRNAs) in the human developing gonad. Using multiple genetic and molecular biology strategies in an in vitro system of female germ cell differentiation, the study further provides compelling evidence of a positive regulatory function of the LNC1845 lncRNA on its protein-coding neighbor LHX8, known to have a role in ovarian follicle development. This study has important interest to reproductive biologists and to the non-coding RNA community.

## Introduction

Germ cells undergo many unique cellular and epigenetic processes during gametogenesis (*Guo et al., 2015*; *Irie et al., 2015*). Recent studies have revealed that some transcriptional factors such as PRDM1

are essential for early primordial germ cell (PGC) specification, while a group of germ cell-specific RNA-binding proteins, including VASA and DAZL, play important roles at later stages of germ cell development (*Kurimoto and Saitou, 2018*; *Li et al., 2020*). Besides coding genes, non-coding RNAs may also directly participate in the developmental process of germ cells. For instance, Piwi-interacting RNAs are a class of non-coding RNAs specifically expressed in germ cells and are required for spermatogenesis (*Aravin et al., 2006*; *Girard et al., 2006*; *Grivna et al., 2006*).

There is growing evidence that diverse classes of RNAs, ranging from short RNAs to long non-coding RNAs (lncRNAs), have emerged as key regulators of gene expression and genome stability (*Krol et al., 2010*; *Perry and Ulitsky, 2016*; *Tam et al., 2008*). The development of next-generation sequencing and algorithmic analysis reveals that more than 100,000 lncRNAs exist in human cells (*Fang et al., 2018*; *Volders et al., 2019*). Notably, over 67% of these lncRNAs are unannotated and not included in the Gencode database, although some might be included in the NONCODE database. Among these lncRNAs, only a small portion of lncRNAs have been functionally characterized, especially in the context of human germ cell (hGC) development. Recently, several reports have shown that lncRNAs play critical roles in *Drosophila* and mouse spermatogenesis and fertility (*Wen et al., 2016*; *Wichman et al., 2017*). However, the expression profiles of lncRNAs and their potential function in hGCs remain to be elucidated. To gain a comprehensive picture of lncRNA functionality in hGC development, it is necessary to identify germline-specific lncRNAs and construct the global lncRNA expression landscape during hGC development.

A published analysis of single-cell transcriptome data of hGCs and gonadal somatic cells (*Li et al., 2017*) was utilized to identify a dynamic expression pattern of lncRNAs during hGC and gonadal somatic cell development. Computational and experimental verification further examined the molecular function of a newly characterized lncRNA, *LNC1845*, and its neighboring gene, *LHX8. LHX8* encodes an LIM homeodomain transcriptional regulator that is preferentially expressed in germ cells and is critical for mammalian oogenesis (*Choi et al., 2008*; *Pangas et al., 2006*). *LNC1845* was found to regulate *LHX8* expression in *cis* by modulating histone modifications in the regions near the *LHX8* transcription start site. Moreover, FOXP3, one of forkhead factors which plays important roles during regulatory T cells differentiation (*Gavin et al., 2007*; *Weigel et al., 1989*), is identified as a major transcriptional activator of lncRNAs in hGCs.

## Results

### Numerous lncRNAs are differentially expressed during hGC development

The expression of lncRNAs during human gonadal development was analyzed using recently reported single-cell transcriptomes (*Li et al., 2017*). There were 1825 single cells involving 1369 PGCs and 456 gonadal somatic cells analyzed from the polyA + transcriptome dataset. The lncRNAs and protein-coding genes were included if a transcript was detected more than 1 transcripts per million reads (TPM) and mapped in at least two samples. The analysis showed that the total number of expressed lncRNAs and the ratio of lncRNAs versus coding genes in germ cells are higher than that in somatic cells of the same gonads (*Figure 1A*). In female and male germ cells (fGCs and mGCs), there were 10,624 and 11,977 expressed lncRNAs, respectively, compared with 4295 and 3186 expressed lncRNAs in female and male gonadal somatic cells (fSOMAs and mSOMAs), respectively. Notably, over 67% of these lncRNAs are unannotated in both Refseq (NCBI) and GENCODE, indicating that only a small proportion of lncRNAs expressed in human gonads have been identified or fully annotated in previous studies. Among these unannotated lncRNAs, some neighboring protein-coding genes are related to spermatogenesis (SPATC1) and the bone morphogenetic protein (BMP) signaling pathway (SMAD1) (*Figure 1B*). The average number of expressed lncRNAs in each germ cell is higher than that in somatic cells from 5 to 26 weeks of gestation (WG) in female gonads and 4–25 WG in male gonads (*Figure 1C*, *Figure 1—figure supplement 1A*, and *Supplementary file 1*). Interestingly, the average number of expressed lncRNAs in each fGC increases to the highest level at around 23 WG, and in each mGC at around 19 WG. In contrast, the average number of lncRNAs in each somatic cell remains at the same level throughout similar periods of gonadal development.

The individual germ cells in the same gonads are not entirely synchronized, so they may undergo different cellular programs. Each germ cell was assigned into its corresponding developmental stage

Chromosomes and Gene Expression | Developmental Biology

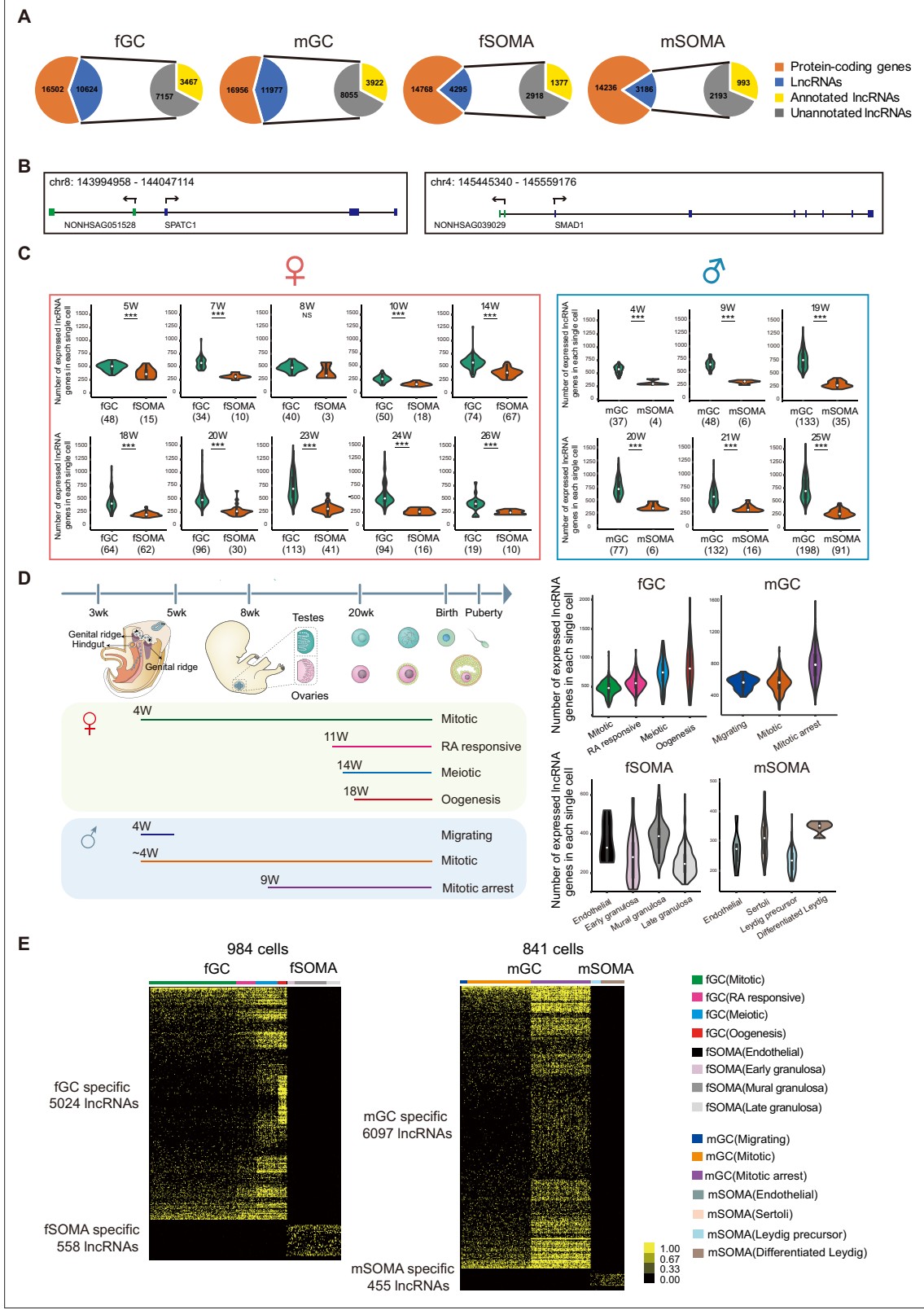

**Figure 1.** Figure numerous. (**A**) Long non-coding RNAs (lncRNAs) are differentially expressed during human germ cell (hGC) development. (**B**) Two unannotated lncRNAs and their neighboring protein-coding genes with their genomic positions. (**C**) Number of lncRNAs in an individual cell of hGCs and gonadal somatic cells from different developmental weeks. Student's t-test was used for the comparisons. ***p < 0.001. (**D**) The composition of different cell types from different developmental weeks and the numbers of lncRNAs in an individual cell from different cell types. The color lines

*Figure 1 continued*

below indicate cell collections from different developmental weeks. (**E**) Expression heatmap of hGC-specific lncRNAs and gonadal somatic cell-specific lncRNAs.

The online version of this article includes the following figure supplement(s) for figure 1:

**Figure supplement 1.** Numerous long non-coding RNAs (lncRNAs) are differentially expressed during human germ cell development.

(fGCs into mitotic, RA responsive, meiotic, and oogenesis; and mGCs into migrating, mitotic, and mitotic arrest) according to the transcriptomes of the coding genes and analyzed the dynamics of lncRNA expression as in a previous study (*Figure 1D*; *Li et al., 2017*). The number of expressed lncRNAs increases from the mitotic to oogenesis stage in fGCs and from migrating to the mitotic arrest stage in mGCs (*Figure 1D*). There is strong heterogeneity in lncRNA number when female hGCs enter meiosis and male hGCs enter mitotic arrest. LncRNA showed dynamic expression pattern although the number of expressed lncRNAs is generally less in gonadal somatic cells. In fSOMA, the lncRNA number increases when cells developed from early granulosa to mural granulosa cells and then decreases in late granulosa cells. In mSOMA, the lncRNA number is higher in differentiated Leydig cells than that in Leydig precursor cells (*Figure 1D*). The trend of expressed protein-coding gene number is similar to lncRNA number, but exhibits more subtle differences. For example, the lncRNA number increases from RA responsive to meiosis stage in fGC, but the protein-coding gene number decreases (*Figure 1D* and *Figure 1—figure supplement 1B*).

To characterize the specificity of lncRNA expression during human gonadal development, a systematically bioinformatical analysis reveals the expression pattern of lncRNAs in specific cell types and developmental stages. A lncRNA is expressed specifically in one cell type or developmental stage with TPM equal or more than one in at least two cells, and the same lncRNA is not expressed in all other cell type with TPM equal or more than one. According to the above criterion, 5024 fGC-, 558 fSOMA-, 6097 mGC-, and 455 mSOMA-specific lncRNAs were identified (*Figure 1E* and *Supplementary file 2*). After clustering the cells into different developmental stages and analyzing the lncRNAs specific to each stage, 1218 and 2942 stage-specific lncRNAs in female and male hGCs were identified, respectively (*Figure 1—figure supplement 1C* and *Supplementary file 3*).

## Integrating lncRNA expression enhances cell-type classifications of human fGCs

Previous analysis showed that human female gonadal germ cell could be clustered into four developmental stage according to protein-coding genes (*Li et al., 2017*). We tested whether combining lncRNA expression with coding gene expression could better define the classification of cell types. As there are many stage-specific lncRNAs expressed, a t-distributed stochastic neighbor embedding (t-SNE) analysis identified cell types of female gonadal germ cells and somatic cells by integrating lncRNAs with protein-coding genes. t-SNE analysis using lncRNAs and coding genes identified an additional cell population clustered close to the mitotic population (*Figure 2A*). As this population clustered closely with three other cell populations which belong to mitotic stage and separate from other cell types, we named this population mitotic-4. This population of cells expressed many genes related to 'protein modification', 'transcription regulation', and 'RNA metabolic' according to the gene ontology (GO) analysis (*Figure 2—figure supplement 1A*).

Using pseudotime trajectory analysis to dissect the developmental stage of the fGCs, mitotic cells are distributed at one end, followed by RA responsive cells, meiotic cells, and oogenesis cells at the opposite end (*Figure 2B* and *Figure 2—figure supplement 1B*). If lncRNA dataset was integrated into the analysis, four mitotic populations can be distinguished along the pseudotime trajectory, whereas only three populations of mitotic cells are distributed according to protein-coding genes. Interestingly, many mitotic-4 cells are located next to the RA-responsive cells and away from the starting point of the trajectory, suggesting that the developmental stage of this population is later than the other three mitotic populations (*Figure 2B*).

Usually, protein-coding genes specifically expressed in certain cell type have been selected as gene markers for cellular and molecular studies. Computational analysis shows that lncRNAs might also be used as markers for specific cell types in female gonadal cells. The top 5 expressed lncRNAs and protein-coding genes of each tSNE cluster were shown in a heatmap (*Figure 2—figure supplement*

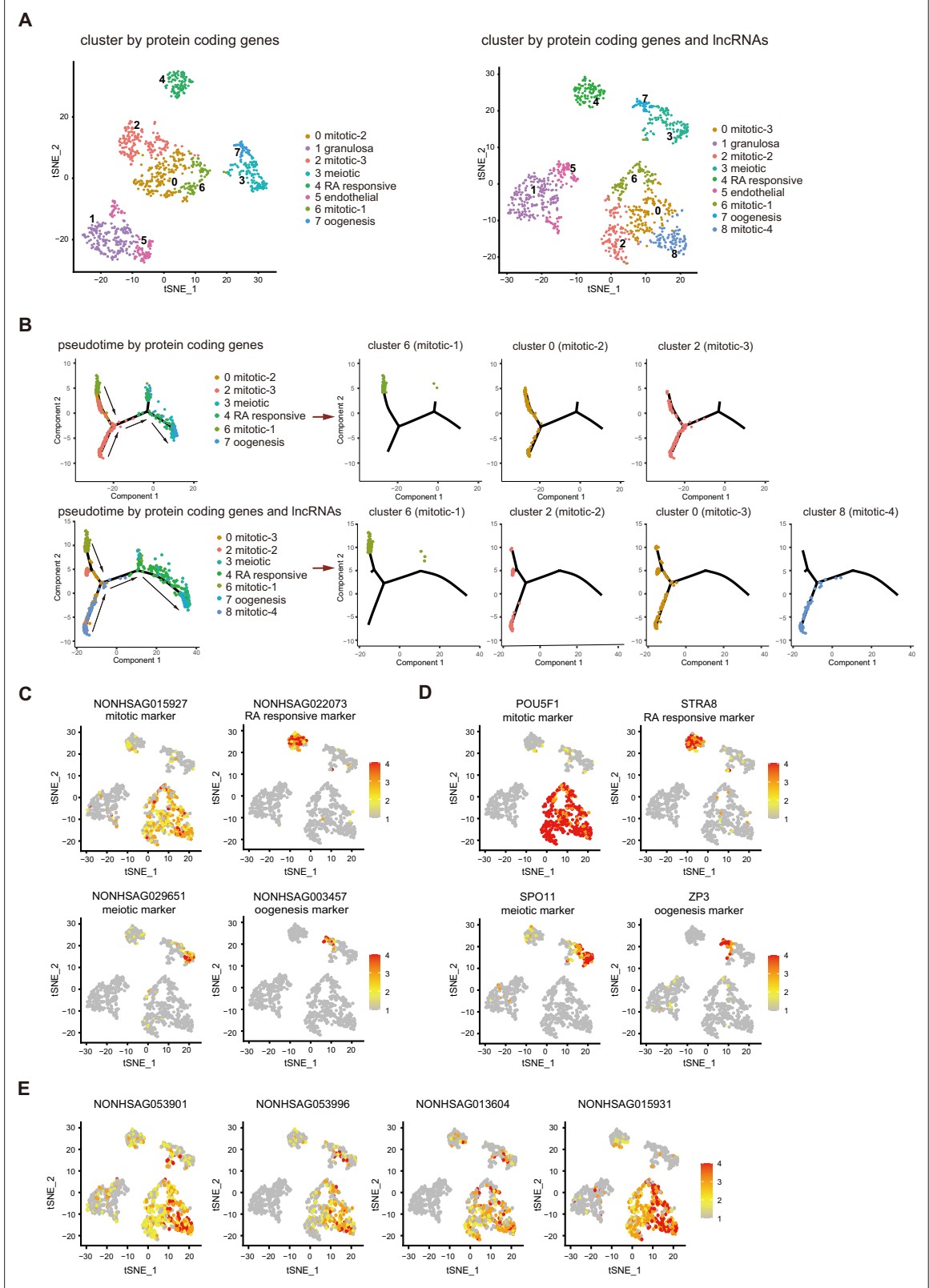

**Figure 2.** Integrating long non-coding RNA (lncRNA) expression enhances cell-type classifications of human female germ cells. (**A**) tSNE plot of female germ cells and somatic cells colored by identified cell types. Left: cluster according to protein-coding genes. Right: cluster according to protein-coding genes and lncRNAs. (**B**) Single-cell trajectories of female germ cell states through the pseudotime according to protein-coding genes or protein-coding genes with lncRNAs. The different subtypes of mitotic stage germ cells are shown through the pseudotime separately. Arrows indicate

*Figure 2 continued on next page*

*Figure 2 continued*

the developmental order of these cells. (**C**) Expression pattern of identified cell-type marker lncRNAs exhibited on t-SNE plots, including mitotic, RA responsive, meiotic, and oogenesis markers. (**D**) Expression pattern of identified cell-type marker coding genes exhibited on t-SNE plots, including mitotic, RA responsive, meiotic, and oogenesis markers. (**E**) Expression pattern of newly identified mitotic-4 cell lncRNA markers exhibited on t-SNE plots. Data information: In (**C–E**), a gradient of gray, yellow, orange, and red indicates low to high expression.

The online version of this article includes the following figure supplement(s) for figure 2:

**Figure supplement 1.** Integrating long non-coding RNA (lncRNA) expression enhances cell-type classifications of human female germ cells.

*1C*). Both protein-coding genes and lncRNAs were selected as specific marker using Seurat FindAll-Marker function (*Supplementary file 4*). Thus, lncRNA markers can be used alone or together with the protein-coding markers in the corresponding cell types (*Figure 2C, D*). Several lncRNAs are more specifically expressed in the newly identified mitotic-4 population (*Figure 2E*), which suggests that cell-type-specific expression of lncRNAs contributes to the new cell-type classification.

## Genomic distributions and biotypes of the lncRNAs expressed during human gonadal development

Genomic distributions of lncRNAs may determine their molecular and cellular functions (*Canzio et al., 2019*; *Luo et al., 2016*; *Subhash et al., 2018*). Among them, approximately 53% and 52% of lncRNAs are intragenic in fGCs and mGCs, respectively, whereas approximately 62% of lncRNAs are intragenic in both fSOMAs and mSOMAs (*Figure 3A*). The subregional distribution of lncRNAs has a similar pattern among the 3′UTR, intron, exon, TSS, ncRNA, 5′UTR, promoter, pseudogene, snoRNA, and miRNA regions in both germ cells and somatic cells (*Figure 3A*). To further assess the differences between annotated and unannotated lncRNA genomic distributions, these lncRNAs were further classified based on their annotations. Genomic distribution analysis showed that unannotated lncRNAs have a higher proportion of intergenic regions compared to annotated lncRNAs (*Figure 3—figure supplement 1A*).

We classified all lncRNAs specifically expressed at different developmental stages of germ cells into six locus biotypes according to previous studies (*Luo et al., 2016*), including head-to-head antisense lncRNA–mRNA pairs (XH), tail-to-tail position (XT), sense lncRNAs downstream (SD) or upstream (SU) of protein-coding genes, and antisense lncRNAs located within (XI) or encompassing a protein-coding gene (XO) (*Figure 3B* and *Supplementary file 5*). The proportions of specific biotypes varied from one developmental stage to another (the number of lncRNAs specifically expressed during the RA-responsive stage of fGCs and the migrating stage of mGCs were too less to calculate the proportions). For instance, the percentage of XH in fGCs is 16.2%, 34.3%, and 14.5% during mitotic, meiotic, and oogenesis stages, respectively, whereas that in mGC is 13.7% and 29.8% during mitotic and mitotic arrest stages, respectively. Nevertheless, the XH biotype occupy the highest percentage in both fGCs and mGCs.

Since lncRNAs may directly regulate the expression of their neighboring genes (*Kopp and Mendell, 2018*), we then calculated the expression correlation of lncRNAs and their neighboring protein-coding genes. In fGCs, mitotic-XH, mitotic-XT, meiotic-XH, oogenesis-XT, and oogenesis-XI show positive correlations, whereas in mGCs, mitotic-XT, mitotic arrest-XH, and mitotic arrest-XO showed positive correlations (*Figure 3C*). In contrast, neither fSOMA nor mSOMA biotypes shows positive correlation (*Figure 3—figure supplement 1B*). By comparing the meiotic-XH pairs and randomly selected XH pairs from the same chromosome or lncRNA and protein-coding gene from different chromosomes, we found the Pearson correlation is higher in meiotic expressed divergent genes compared with the other two groups (*Figure 3—figure supplement 1C*). GO analysis revealed that the genes associated with divergent lncRNAs (XH) are enriched in reproductive structure development, transcription regulation, sex differentiation**,** and secretion (*Figure 3—figure supplement 1D*). A positive correlation between lncRNAs and their neighboring genes suggests that the lncRNAs might positively regulate the expression of neighboring genes.

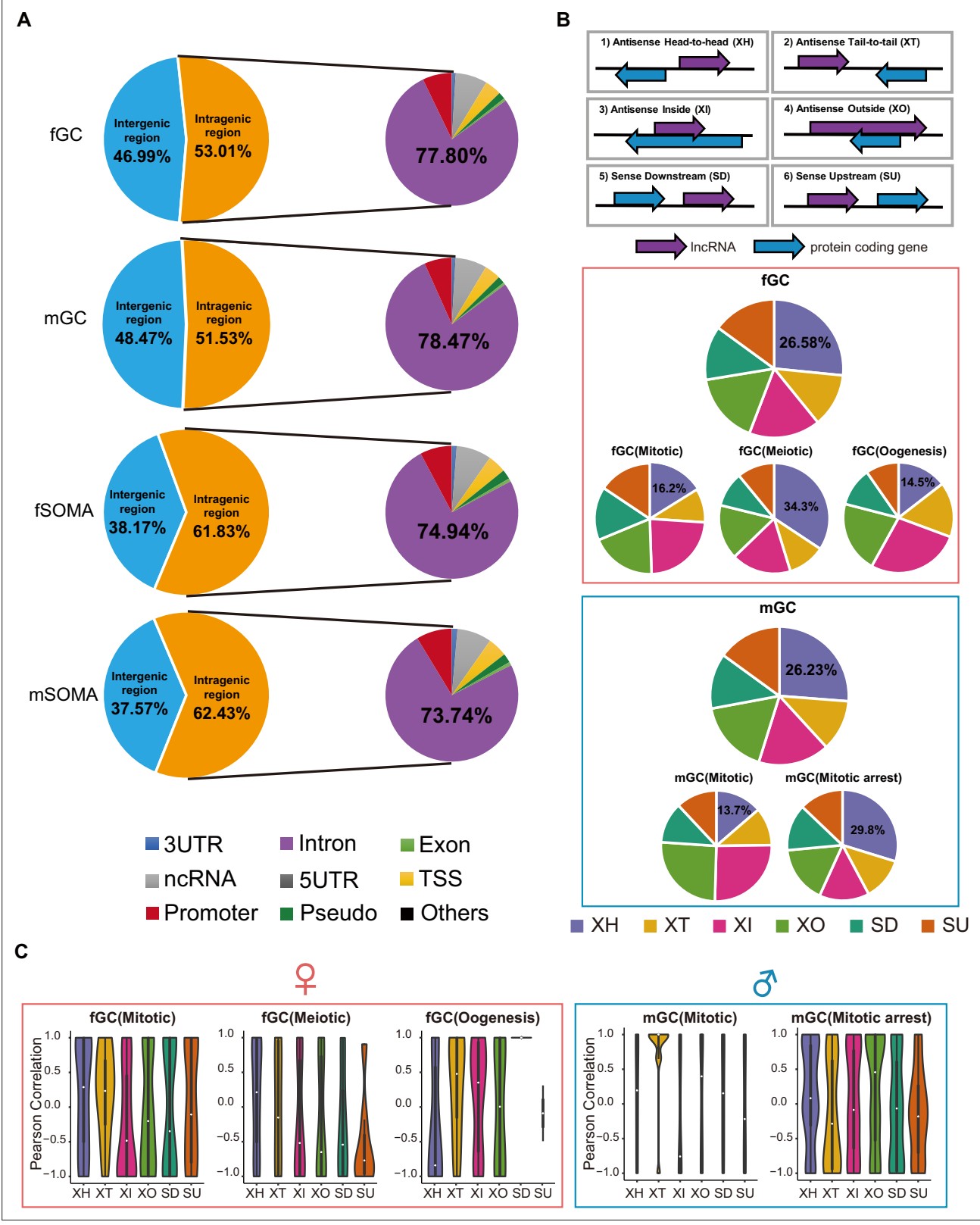

**Figure 3.** Genomic distributions and biotypes of the long non-coding RNAs (lncRNAs) expressed during human gonadal development. (**A**) The genomic position distribution of lncRNAs expressed in human germ cells (hGCs) and gonadal somatic cells. (**B**) The distribution and percentage of the six locus biotypes of each developmental stage-specific lncRNAs. (**C**) Expression correlations of lncRNA–mRNA pairs of the six biotypes in different developmental stages.

*Figure 3 continued on next page*

*Figure 3 continued*

The online version of this article includes the following figure supplement(s) for figure 3:

**Figure supplement 1.** Genomic distributions and biotypes of the long non-coding RNAs (lncRNAs) expressed during human gonadal development.

## Characterization of a divergent lncRNA, *LNC1845*, expressed during the female meiotic stage

Among these divergent lncRNA–mRNA pairs, we found 11 XH pairs are mainly expressed in meiotic cells and show positive correlations (*Figure 3—figure supplement 1E*). To validate the regulatory potential of the lncRNAs, we selected the lncRNA and the neighboring gene by considering the following criteria: (1) lncRNA expression is stage specific; (2) the neighboring protein-coding gene has reported function in meiosis or oogenesis; and (3) gene expressions of both lncRNA and the coding gene are detectable in an in vitro differentiation system of human embryonic stem cells (hESCs) to primordial germ cell-like cells (PGCLCs). Among the 11 XH pairs, *NONHSAG001845* (*LNC1845*) and *LHX8* fulfill the first two criteria, so we next examined whether *LNC1845* on *LHX8* are both expressed in the in vitro differentiated PGCLCs and the characterizations of *LNC1845*.

According to the NONCODE and GENCODE database and our RNA-seq result of PGCLCs, *LNC1845* transcription yields a 1410 nucleotide, nonspliced, and polyadenylated lncRNA that was initiated 4507 bases upstream of *LHX8* (*Figure 4A*). According to protein-coding prediction programs Coding Potential Calculator and Coding Potential Assessment Tool (*Kong et al., 2007*; *Wang et al., 2013*), *LNC1845* is most likely a non-coding transcript (*Figure 4—figure supplement 1A*). We validated the non-coding property of *LNC1845* by an in vitro transcription and translation system and confirmed that no protein product was detected using *LNC1845* as the gene template (*Figure 4—figure supplement 1B*; *Figure 4—figure supplement 1—source data 1*). The physiological expression of *LNC1845* was confirmed by performing reverse transcription-PCR analysis using total RNA from 17 WG human fetal ovaries. The transcriptional expression of *LNC1845* was readily detected in the 17 WG human fetal ovary samples but was not detected in human ovary carcinoma; it was very low in undifferentiated hESCs H9 (*Figure 4B*). These results were consistent with the single-cell transcriptome analysis reported previously showing specific expression of *LNC1845* in fGCs of 17 WG fetal ovaries (*Figure 4C* and *Figure 4—figure supplement 1C*; *Guo et al., 2015*; *Li et al., 2017*). *LHX8* expression was also detected in the fGCs in the same period of fGCs and detected in more developmental stages. According to GENECODE, *LNC1845* has only one isoform and no intron. We also validated its RNA transcript and sequence by RACE (*Figure 4—figure supplement 1D*; *Figure 4—figure supplement 1—source data 1*).

To examine the molecular function of *LNC1845*, we required an in vitro cellular system containing germ cells expressing *LNC1845* and its neighboring gene, *LHX8*. Previous studies have shown that in vitro differentiation of hESCs into late PGCLCs may provide the appropriate cell types that match the developmental stages to study the molecular function of *LNC1845* (*Jung et al., 2017*; *Kee et al., 2009*; *Panula et al., 2011*). BMPs and DAZL were used to induce late PGCLCs, and these cells were collected by FACS for further analysis (*Figure 4D*). A total of 397 germ cell-specific coding genes, including *LHX8* and 117 germ cell-specific lncRNAs, including *LNC1845*, that we identified in the single-cell transcriptomes were expressed in the late PGCLCs (*Figure 4E, F* and *Figure 4—figure supplement 1E, F*, *Supplementary file 6* and *Supplementary file 7*). After analyzing the co-expression pattern of XH lncRNA and their neighboring coding genes, 36 lncRNA-coding gene pairs, including *LNC1845* and *LHX8* were both enriched in the late PGCLCs but not in the differentiated control cells and undifferentiated hESCs (*Figure 4—figure supplement 2*). qPCR analysis also confirmed that *LNC1845* and *LHX8* enrichment in PGCLCs but not the control cells (*Figure 4G*). Moreover, immunostaining and RNA fluorescent in situ hybridization (RNA FISH) experiments show that LHX8 and *LNC1845* colocalized in the same nuclei of PGCLCs but not in the control cells (*Figure 4H* and *Figure 4—figure supplement 1G*).

Among the 36 lncRNA-coding gene pairs, *LNC1845* and *LHX8* satisfied all three criteria of our selection as a stage-specific lncRNA, the coding gene has reported function in oogenesis, and both expressed in the PGCLCs. Hence, we focused on studying the potential regulatory mechanism of *LNC1845–LHX8* pair.

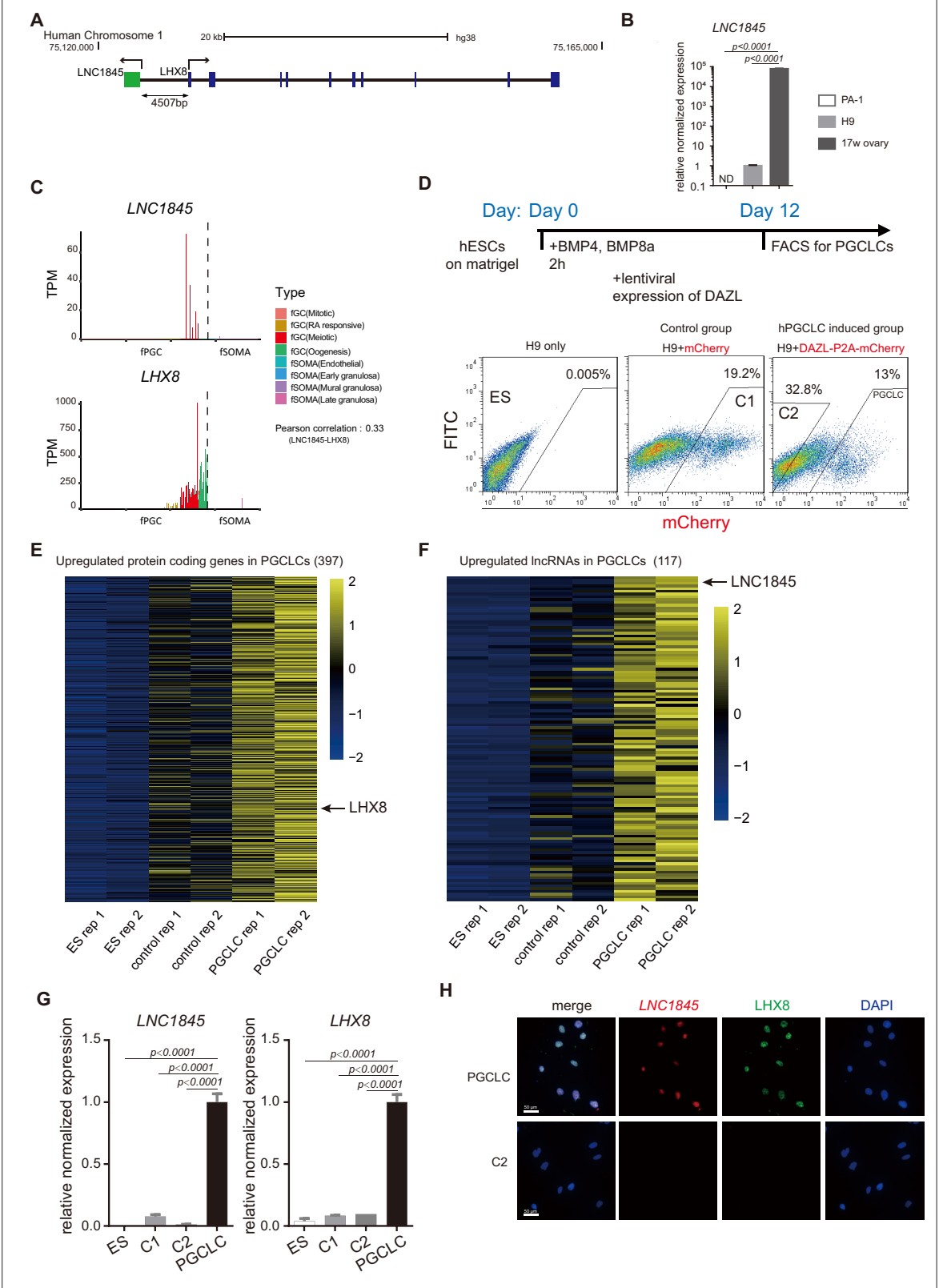

**Figure 4.** Characterization of a divergent long non-coding RNA (lncRNA), *LNC1845*, expressed during the meiotic stage and is required for normal expression of *LHX8*. (**A**) Genomic locus and relative position of *LNC1845* (in green) and *LHX8* (in blue). (**B**) Expression analysis by RT-qPCR of PA-1 cells, undifferentiated H9 cells, and RNA sample extracted from human 17-week ovary (*n* = 3 technical replicates). (**C**) *LNC1845* and *LHX8* expression level in single hPGC or gonadal somatic cells from different developmental stages. (**D**) Schematic timelines for primordial germ cell-like cell (PGCLC) induction.

*Figure 4 continued on next page*

*Figure 4 continued*

FACS plots showing distinct populations expressing mCherry fluorescent proteins, indicating control groups or PGCLCs. (**E**) Expression heatmap of female germ cell (fGC)-specific protein-coding genes upregulated in PGCLCs. (**F**) Expression heatmap of fGC-specific lncRNAs upregulated in PGCLCs. (**G**) Expression analysis by RT-qPCR of undifferentiated H9 (ES) cells (*n* = 3), mCherry-positive control (control-1) cells (*n* = 3), DAZL-negative (control-2) cells (*n* = 3), and DAZL-positive PGCLCs. (**H**) Immunofluorescence of LHX8 (green) co-stained with RNA fluorescent in situ hybridization (RNA FISH) of *LNC1845* (red) in PGCLCs or control cells. Scale Bar, 50 μm. In (**B**) and (**G**), the *y*-axis represents relative mean expression normalized to *GADPH* and control cells. Error bars indicate the mean ± standard deviation (SD), three independent experiments were carried out for (**G**). One-way analysis of variance (ANOVA) was used for the comparisons.

The online version of this article includes the following source data and figure supplement(s) for figure 4:

**Figure supplement 1.** Characterization of a divergent long non-coding RNA (lncRNA), *LNC1845*, expressed during the meiotic stage and is required for normal expression of *LHX8*.

**Figure supplement 1—source data 1.** Characterization of a divergent long non-coding RNA (lncRNA), *LNC1845*, expressed during the meiotic stage and is required for normal expression of *LHX8*.

**Figure supplement 2.** Upregulated XH long non-coding RNA (lncRNA)–mRNA pairs in hPGCLCs.

### *LNC1845* RNA transcripts are required for normal expression of *LHX8*

To test the hypothesis that *LHX8* transcription is regulated by *LNC1845*, we firstly deleted *LNC1845* through gene targeting in the hESC H9 line. As lncRNAs often play regulatory roles by complementary base pairing (*Carrieri et al., 2012*; *Ebert et al., 2007*; *Gong and Maquat, 2011*), partial deletion or coding frameshift editing cannot completely eliminate the function of lncRNAs. Hence, we deleted the whole *LNC1845* using the CRISPR/Cas9 gene targeting system with a two-step deletion strategy. First, we used a pair of gRNAs targeting near 5′ of LNC1845 locus and inserted drug selection marker through homologous recombination. After drug selection for single colonies, we used Cre system to remove drug selection sequences. We confirmed the deletion by PCR tests and sequencing of the targeted region (*Figure 5A* and *Figure 5—figure supplement 1A*; *Figure 5—figure supplement 1—source data 1*, *Figure 5—figure supplement 1—source data 2*). In the *LNC1845* KO cell lines differentiated to late PGCLCs, the RNA transcripts of *LNC1845* were not detectable by qPCR analysis but were readily detectable in WT cells (*Figure 5B*). Notably, *LHX8* transcriptional and protein expression were also greatly reduced in *LNC1845* KO cells compared with that in WT cells (*Figure 5B, C*). In addition to *LHX8*, the expression of many genes was also affected by *LNC1845* deletion, including genes that participate in cell motility, cell communication, and cell proliferation, although the effects might be direct or indirect (*Figure 5D* and *Figure 5—figure supplement 1B*, *Supplementary file 8*). To further dissect *LNC1845* function beyond regulating LHX8, we examined how similar are the gene expressions affected by LNC1845 and the gene expressions affected by overexpression of LHX8 in PGCLCs (LHX8 OE). Thus, we compared DEGs of *LNC1845* KO and LHX8 OE cells separately, and analyzed whether the majority of these DEGs were overlapped. As the RNA-seq data show, there are 1126 DEGs overlapped between these two groups, and many meiotic related genes are among them (*Figure 5—figure supplement 1C* and *Supplementary file 9*). We also found some meiotic related genes and apoptosis related genes are differentially expressed when *LNC1845* KO but not in LHX8 OE group, which indicating that these regulations may directed by *LNC1845* rather than LHX8 (*Figure 5—figure supplement 1C*).

As *LHX8* has two isoforms, we also tested which isoform is dominant and whether it is regulated by *LNC1845*. We designed transcript variant-specific primers to detect the expression of two isoforms and found both isoforms are expressed in the in vitro PGCLC induction system and both are regulated by *LNC1845* (*Figure 5—figure supplement 1D*).

We next tested whether the RNAi-mediated downregulation of *LNC1845* expression might also alter *LHX8* expression levels without modifying the DNA sequence or chromosomal structure proximal to *LHX8*. After obtaining two shRNAs that efficiently knocked down *LNC1845* in the late PGCLCs, we found that *LHX8* RNA and protein expression were simultaneously reduced in the same cells compared with that in the control cells infected with sh*LACZ* (*Figure 5C, E*; *Figure 5—source data 1*, *Figure 5—source data 2*).

We further validated the regulation of *LHX8* by *LNC1845* by inserting transcription stop signals to inhibit lncRNA transcription or destabilize its RNA product, which is commonly used to study the function of lncRNAs (*Gutschner et al., 2011*). We knocked in the 3×polyA stop cassette immediately downstream of the transcription start site of *LNC1845* via targeted homologous recombination using

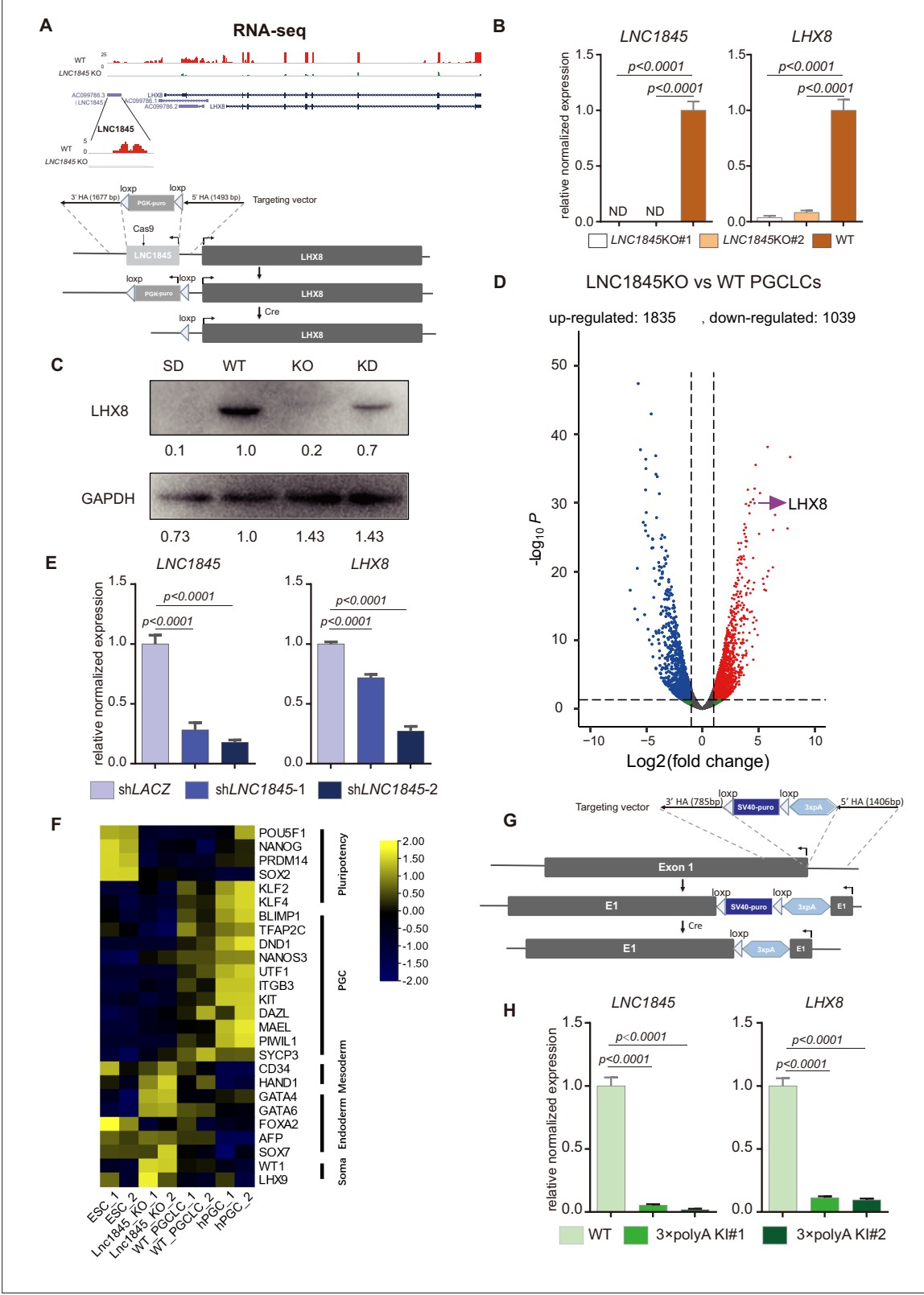

**Figure 5.** *LNC1845* RNA transcripts are required for normal expression of *LHX8*. (**A**) Schematic diagram of knockout strategies at the *LNC1845* loci. (**B**) Expression analysis by RT-qPCR of WT primordial germ cell-like cells (PGCLCs) (*n* = 3) and *LNC1845* KO PGCLCs (*n* = 3). (**C**) Western blot analysis of LHX8 in PGCLCs from WT, *LNC1845* KO, or RNA interfering *LNC1845* (KD) human embryonic stem cells (hESCs). SD, spontaneous differentiation cells as the negative control. (**D**) Volcano plots of the differentially expressed protein-coding genes between *LNC1845*KO and WT PGCLCs. The screening

*Figure 5 continued on next page*

*Figure 5 continued*

threshold of differentially expressed genes (fold change) is ≥2, p < 0.05. (**E**) Expression analysis by RT-qPCR analysis of *LNC1845* and *LHX8* in *LNC1845* RNAi cells (*n* = 3). (**F**) Heatmap of gene expression of key primordial germ cell (PGC)-associated genes and of pluripotency, mesoderm, endoderm, and somatic markers. (**G**) Schematic diagram of *LNC1845*-upstream 3×polyA knock-in and the targeted allele after 3×polyA insertion. (**H**) Expression analysis by RT-qPCR of *LNC1845* and *LHX8* in WT PGCLCs (*n* = 3) and *LNC1845* 3×polyA KI cells (*n* = 3). In (**B**), (**E**), and (**H**), the *y*-axis represents relative mean expression normalized to *GADPH* and control cells. Error bars indicate the mean ± standard deviation (SD), three independent experiments were carried out. One-way analysis of variance (ANOVA) was used for the comparisons.

The online version of this article includes the following source data and figure supplement(s) for figure 5:

**Source data 1.** Western blot analysis of LHX8 in PGCLCs from WT, *LNC1845* KO, or RNA interfering *LNC1845* (KD) hESCs. SD, spontaneous differentiation cells as the negative control.

**Source data 2.** Western blot analysis of GAPDH in PGCLCs from WT, *LNC1845* KO, or RNA interfering *LNC1845* (KD) hESCs. SD, spontaneous differentiation cells as the negative control.

**Figure supplement 1.** *LNC1845* RNA transcripts are required for normal expression of *LHX8*.

**Figure supplement 1—source data 1.** Genotyping of *LNC1845* KO single colonies to confirm the deletion of *LNC1845*.

**Figure supplement 1—source data 2.** Genotyping of *LNC1845*-upstream 3×polyA insertion single colonies.

the CRISPR/Cas9 system (*Figure 5F*). A total of 11 homozygous insertion cell lines were created (*Figure 5—figure supplement 1E*; *Figure 5—figure supplement 1—source data 1*, *Figure 5—figure supplement 1—source data 2*), and we used two cell lines (L1 and L12) for further analysis. Quantitative PCR analysis showed that 3×polyA insertion results in an 80–90% decrease in *LNC1845* transcripts (*Figure 5G*). Consistent with the *LNC1845* KO result, *LHX8* expression is also significantly reduced in *LNC1845* + polyA cells as compared with that in the WT PGCLCs (*Figure 5G*). Taken together, several orthogonal experiments reveal that *LNC1845* transcriptional expression is required for proper *LHX8* expression in late PGCLCs.

## *LNC1845* regulates *LHX8* expression in *cis* by modulating chromatin modifications

Previous studies have shown that some lncRNAs may exert *cis*-activating regulations to impact the transcriptional expression of their neighboring genes (*Gil and Ulitsky, 2020*). The physical proximity of *LNC1845* and *LHX8* suggests that this type of regulation may occur at this locus. First, we tested whether *LNC1845* may regulate *LHX8* expression through *cis*-regulation by inserting a constitutive promoter EF1α immediately upstream of the TSS of *LNC1845* (*Figure 6A* and *Figure 6—figure supplement 1A*; *Figure 6—figure supplement 1—source data 1*, *Figure 6—figure supplement 1—source data 2*). In two independent *LNC1845* + EF1α cell lines, *LNC1845* expression is 3.8- and 13.3-fold higher than that in WT in the late PGCLCs, while *LHX8* expression increases to 3.5- and 4.2-fold (*Figure 6B*). Protein expression of *LHX8* also simultaneously increases in the *LNC1845* + EF1α PGCLCs (*Figure 6C*; *Figure 6—source data 1*, *Figure 6—source data 2*). All the cell lines used in the paper generated via CRISPR/Cas9, including *LNC1845* KO, *LNC1845* + 3×polyA and *LNC1845* + EF1α, are karyotypically normal (*Figure 6—figure supplement 1B*).

In addition to inserting a constitutive promoter, we also tested potential *cis*-regulation using the CRISPR-ON system (*Gilbert et al., 2014*). Co-expression of three different sgRNAs targeting the *LNC1845* promoter region with a dCas9-VP64 activator in 293 FT cells increases the *LNC1845* expression level by 1.6- to 4.2-fold, and this activation upregulates *LHX8* expression level by 4- to 7.6-fold (*Figure 6D*). We also targeted the *LNC1845* promoter region in wild-type or *LNC1845* KO ES cells and found in wild-type ES cells, the targeting could increase the *LNC1845* expression level by 3-fold, and *LHX8* expression level was also upregulated by 2.5-fold, but could not activate *LNC1845* or *LHX8* expression in *LNC1845* KO ES cells, indicating that *LHX8* upregulation in WT cells was due to *LNC1845* but not gRNA-activating *LHX8* directly (*Figure 6—figure supplement 1C*). Both CRISPR-ON and EF1α promoter knock-in experiments indicated that the *LNC1845* RNA transcript could *cis*-activate *LHX8* expression.

Conversely, we also tested whether *LNC1845* may *trans*-activate *LHX8* by two independent experimental approaches, namely, transfections of in vitro transcripts and transductions of lentiviral vectors. First, we prepared the *LNC1845* RNA transcript using the T7 in vitro transcription system (*Figure 6—figure supplement 1D*; *Figure 6—figure supplement 1—source data 1*) and transfected

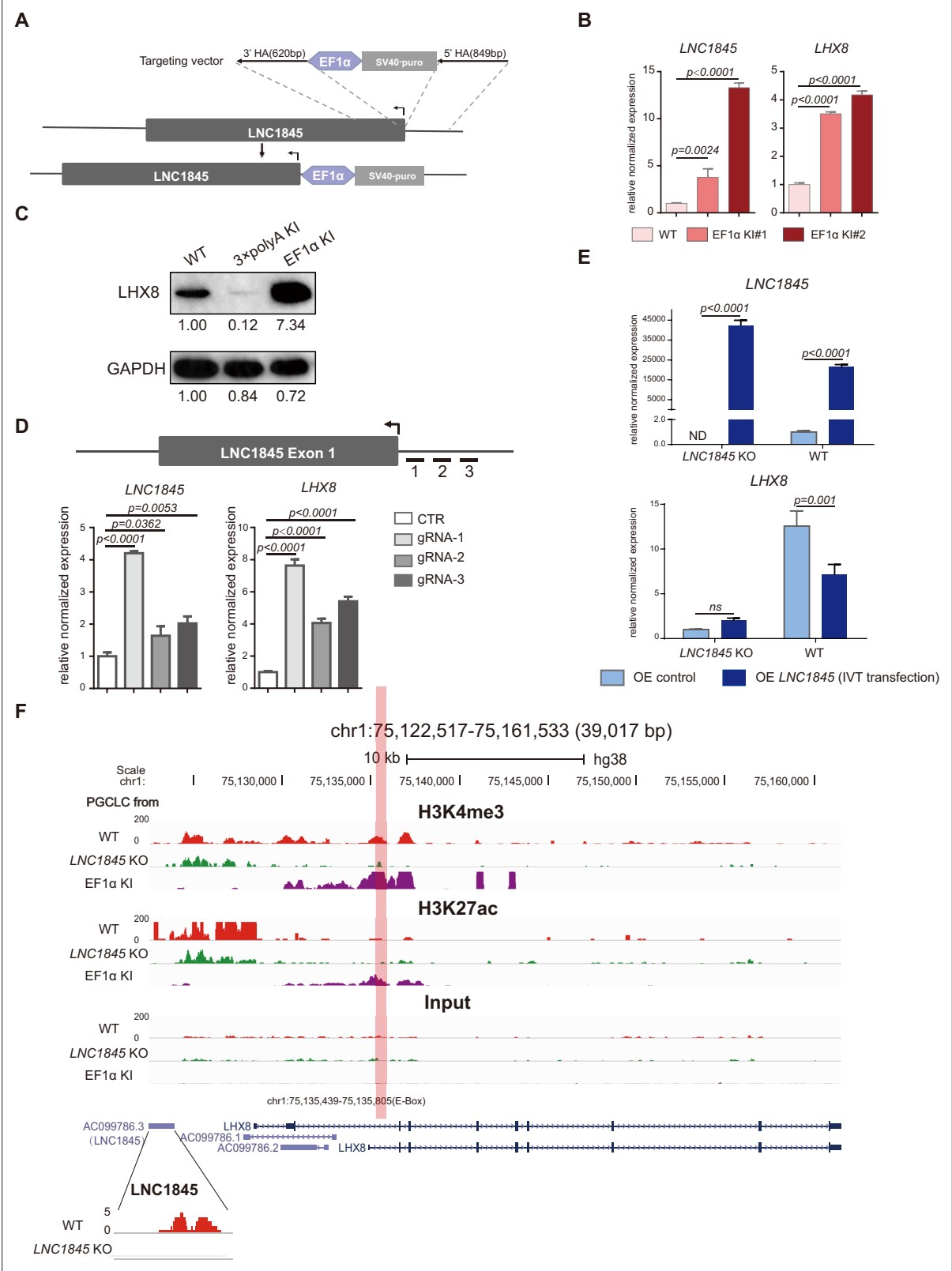

**Figure 6.** *LNC1845* regulates *LHX8* expression in *cis* by changing chromatin modifications. (**A**) Schematic diagram of *LNC1845*-upstream EF1α promoter knock-in and the targeted allele after EF1α promoter insertion. (**B**) Expression analysis by RT-qPCR of *LNC1845* and *LHX8* in WT primordial germ cell-like cells (PGCLCs) (*n* = 3) and *LNC1845*-upstream EF1α promoter knock-in cells (*n* = 3). (**C**) Western blot analysis of *LHX8* in PGCLCs from WT, *LNC1845*-upstream 3×polyA knock-in, or *LNC1845*-upstream EF1α promoter knock-in cells. (**D**) CRISPR-ON-mediated *LNC1845* activation and *LHX8*

*Figure 6 continued on next page*

*Figure 6 continued*

upregulation. RT-qPCR analysis of *LNC1845* and *LHX8*, with or without gRNAs (*n* = 3). Short lines with numbers indicate the relative locations of sgRNAs. (**E**) RT-qPCR analysis of *LNC1845* and *LHX8* in cells overexpressing *LNC1845* transcripts (*n* = 3). (**F**) H3K4me3 and H3K27Ac levels at the *LNC1845/LHX8* locus. These tracks show normalized read densities of H3K4me3 and H3K27Ac in PGCLCs differentiated from WT, *LNC1845* KO, or *LNC1845*-upstream EF1α promoter knock-in human embryonic stem cells (hESCs). The E-BOX region of *LHX8* locus is boxed in orange. Some peaks will exceed the largest scale for better results presenting. In (**B**), (**D**), and (**E**), the *y*-axis represents relative mean expression normalized to *GADPH* and control cells. Error bars indicate the mean ± standard deviation (SD), three independent experiments were carried out. One-way analysis of variance (ANOVA) was used in (**B**) and (**D**), and two-way ANOVA was used in (**E**) for the comparisons.

The online version of this article includes the following source data and figure supplement(s) for figure 6:

**Source data 1.** Western blot analysis of *LHX8* in PGCLCs from WT, *LNC1845*-upstream 3×polyA knock-in, or *LNC1845*-upstream EF1α promoter knock-in cells.

**Source data 2.** Western blot analysis of *GAPDH* in PGCLCs from WT, *LNC1845*-upstream 3×polyA knock-in, or *LNC1845*-upstream EF1α promoter knock-in cells.

**Figure supplement 1.** *LNC1845* regulates *LHX8* expression in *cis* by changing chromatin modifications.

**Figure supplement 1—source data 1.** Genotyping of *LNC1845*-upstream EF1α promoter insertion single colonies.

**Figure supplement 1—source data 2.** Agarose gel analysis of *LNC1845* transcripts by in vitro transcription.

**Figure supplement 2.** *LNC1845* regulates *LHX8* expression in *cis* by physically interacting with WDR5.

**Figure supplement 2—source data 1.** *LNC1845* binding for WDR5 was verified by CHIRP-western blot.

these purified transcripts into WT and *LNC1845* KO hESCs. In the cells transfected with *LNC1845* transcripts, approximately 21,794- and 42,693-fold increases in *LNC1845* transcript are detected in WT and *LNC1845* KO cells, respectively, but *LHX8* expression is only slightly 2-fold higher in KO cells and decreased in WT cells, respectively, compared to the control groups (***Figure 6E***). Second, we also tested the *trans*-activating effect by overexpressing *LNC1845* using a lentiviral system in both WT and *LNC1845* KO PGCLCs with various level of lentiviral concentrations. The levels of *LNC1845* are ~500- to ~40,000-fold higher in *LNC1845* KO and WT cells, respectively, than the control groups, but *LHX8* expression shows no obvious change in KO cells and decreased in WT cells, respectively (***Figure 6—figure supplement 1E***). Therefore, both approaches suggested that *LNC1845* might not have a *trans*-activating effect or that the effect was not significant. To examine the subcellular localizations of LNC1845, cellular fractionation assay was conducted. qPCR results showed that the majority of LNC1845 was detected in nucleus, similar to other lncRNAs such as MALAT1 and NEAT1 (***Figure 6—figure supplement 1F***). In contrast, mRNA of GAPDH was mostly detected in cytoplasm of the same cells.

If *LNC1845* regulates *LHX8* expression through *cis*-activation, histone modifications may be modulated by *LNC1845* (***Kopp and Mendell, 2018***; ***Subhash et al., 2018***). Hence, we examined the levels of two important histone modifications of transcription activation, H3K4me3 and H3K27Ac, in the WT, *LNC1845* KO, and KO, and *LNC1845* + EF1α PGCLCs by the chromatin immunoprecipitation assay ULI-NChIP (***Brind'Amour et al., 2015***). We detected a reduced level of H3K4me3 in *LNC1845* KO cells but an increased level in *LNC1845* + EF1α cells at the predicted promoter region of *LHX8* (***Figure 6F***). This locus contains an E-box sequence previously reported to be the promoter region of *Lhx8* (***Pangas et al., 2006***). Similarly, H3K27Ac levels are increased in *LNC1845* OE cells, although the level of H3K27Ac was very low in WT cells. Taken together, the results indicated that the expression level of *LNC1845* modulates the level of these two histone modifications at the *LHX8* promoter.

The latest discovery of many lncRNAs in association with specific chromatin modification complexes (***Davidovich and Cech, 2015***; ***Guttman et al., 2011***; ***Wysocka et al., 2005***) suggests that *LNC1845* may manage this gene-specific H3K4me3 modification. WDR5 is a highly conserved WD40 repeat protein, and it is the common component of MLL1, MLL2, and hSet1 methyltransferase complex (***Wysocka et al., 2005***). Previous studies have reported the interactions of WDR5 with lncRNA (***Subhash et al., 2018***; ***Wang et al., 2011***; ***Yang et al., 2014***), and the interaction between WDR5 and lncRNAs can guide the methyltransferase to specific locus to establish H3K4me3 status (***Subhash et al., 2018***). Single-cell expression analysis showed WDR5 expresses in both germ cells and somatic cells of human fetal ovaries, but germ cells tend to have higher mRNA expression of WDR5, and the proteins tends to colocalize in hPGCs with high DAZL expression (***Figure 6—figure supplement 2A, B***). In PGCLCs, WDR5 knockdown led to reductions of both *LNC1845* and *LHX8* in the same cells

(*Figure 6—figure supplement 2C*). Thus, WDR5 appears to be critical for *LNC1845* and *LHX8* expressions. Next, we tested whether *LNC1845* RNA may physically interact with WDR5 by performing RNA immunoprecipitation (RIP) in PGCLCs. The results showed that *LNC1845* transcripts were enriched in WDR5 IP sample, similar to the known interacting partner of WDR5, *NR2F2* (*Subhash et al., 2018*; *Figure 6—figure supplement 2D*). Moreover, we performed chromatin isolation by RNA purification (ChIRP) assay using antisense oligos along the entire LNC1845 transcript sequence. ChIRP results confirmed that WDR5 protein were enriched when anti-LNC1845 oligo probes were used to isolate the complex but not the controls without the probes or without overexpression of LNC1845 transcript (*Figure 6—figure supplement 2E*). Taken together, the findings of both approaches support the model that *LNC1845* directly interacts with WDR5 to modulate the H3K4me3 modification for *LHX8* transcriptional activation.

## FOXP3 upregulates expression of *LNC1845* and other lncRNAs

As there are numerous lncRNAs upregulated during fetal germ cell development, we postulated that the transcription expression of these lncRNAs, including *LNC1845*, may be regulated by one or more common transcriptional factors during each developmental stage. To discern the transcription factor(s) that activate lncRNAs, we first formulated a multistep computational prediction method to identify which transcription factor is most likely to bind to the promoter sequences of the lncRNAs during the meiotic stage (*Figure 7—figure supplement 1A*). To look for potential transcription factors, we first used the PROMO program to analyze the promoter sequences of expressed lncRNAs and matched them to the known preferred binding sequences of transcription factors in the database. Next, we selected the candidate genes that were expressed in fGCs for further study. Interestingly, we found the preferred binding motifs of CEBPA, HNF1A, GATA1, TFAP2A, FOXA1, PAX5, STAT4, and FOXP3 are identified in 21–89% of the promoter sequences of lncRNAs expressed in the meiosis stage (385 total meiotic specific lncRNAs). Among them, TFAP2A and FOXP3 are specifically expressed in the in vivo PGCs and in the in vitro PGCLCs (*Figure 7—figure supplement 1B–D*). Promoter of *LNC1845* contains both FOXP3- and TFAP2A-binding sites, so we tested whether overexpression of these transcription factors could upregulate the transcription of *LNC1845*. First, we tested the effect of overexpressing FOXP3 in 293 FT cells. The coding sequence of FOXP3 was inserted into a lentiviral vector with E1Fα promoter (more details in Method) and transduced into 293 FT cells. qPCR analysis showed that overexpression of FOXP3 alone upregulates *LNC1845* and *LHX8* expression, but TFAP2A alone did not show the upregulating effect (*Figure 7—figure supplement 1E*). Second, a modified overexpression vector was conducted in which FOXP3 was cloned and linked with P2A-mCherry in order to collect the overexpressed FOXP3 hPGCLCs by flow cytometry sorting for mCherry-positive cells. FOXP3 overexpressed cells (FOXP3-P) showed higher expression of LNC1845 and LHX8 compared with the control FOXP3-N cells (*Figure 7A*).

Transcriptome analysis of 293 FT cells overexpressing FOXP3 showed that there are 141 meiosis-specific lncRNAs (36.6%) upregulated by FOXP3 and 17 of the lncRNAs (4.4%) are downregulated by overexpression of FOXP3 alone (*Figure 7B*). As the expression level of these lncRNAs is normally lower and the p values used in this analysis are higher than normal protein-coding gene analysis, we further tested 10 of the upregulated lncRNAs identified in the transcriptomes and found that all are significantly upregulated to various levels (*Figure 7—figure supplement 1F*). These results suggest that FOXP3 is a common transcriptional activator of lncRNAs expressed in meiosis stage and *LNC1845* is only one of the activated lncRNAs. Among these upregulated lncRNAs, we found there are many divergent lncRNAs along with their neighboring protein-coding genes which are also upregulated by FOXP3 (*Figure 7C*). To test the hypothesis that these protein-coding genes are also regulated by their neighboring lncRNAs, we knocked down two of these divergent lncRNAs, *NONHSAG003346* and *NONHSAG015266* by two shRNA together with FOXP3 overexpression, and found their neighboring protein-coding genes are simultaneously reduced in the same cells compared with that in the control cells infected with sh*LACZ* (*Figure 7D*). We also tested whether the RNAi-mediated downregulation of FOXP3 might alter *LNC1845* and *LHX8* expression levels. After obtaining two shRNAs that efficiently knocked down FOXP3 in the late PGCLCs, we found that both *LNC1845* and *LHX8* are simultaneously reduced in the same cells compared with that in the control cells infected with sh*LACZ* (*Figure 7E*).

To further test whether FOXP3 regulates lncRNAs by directly binding to the promoter regions, we examined FOXP3-binding activity by measuring luciferase transcription of *LNC1845* promoter.

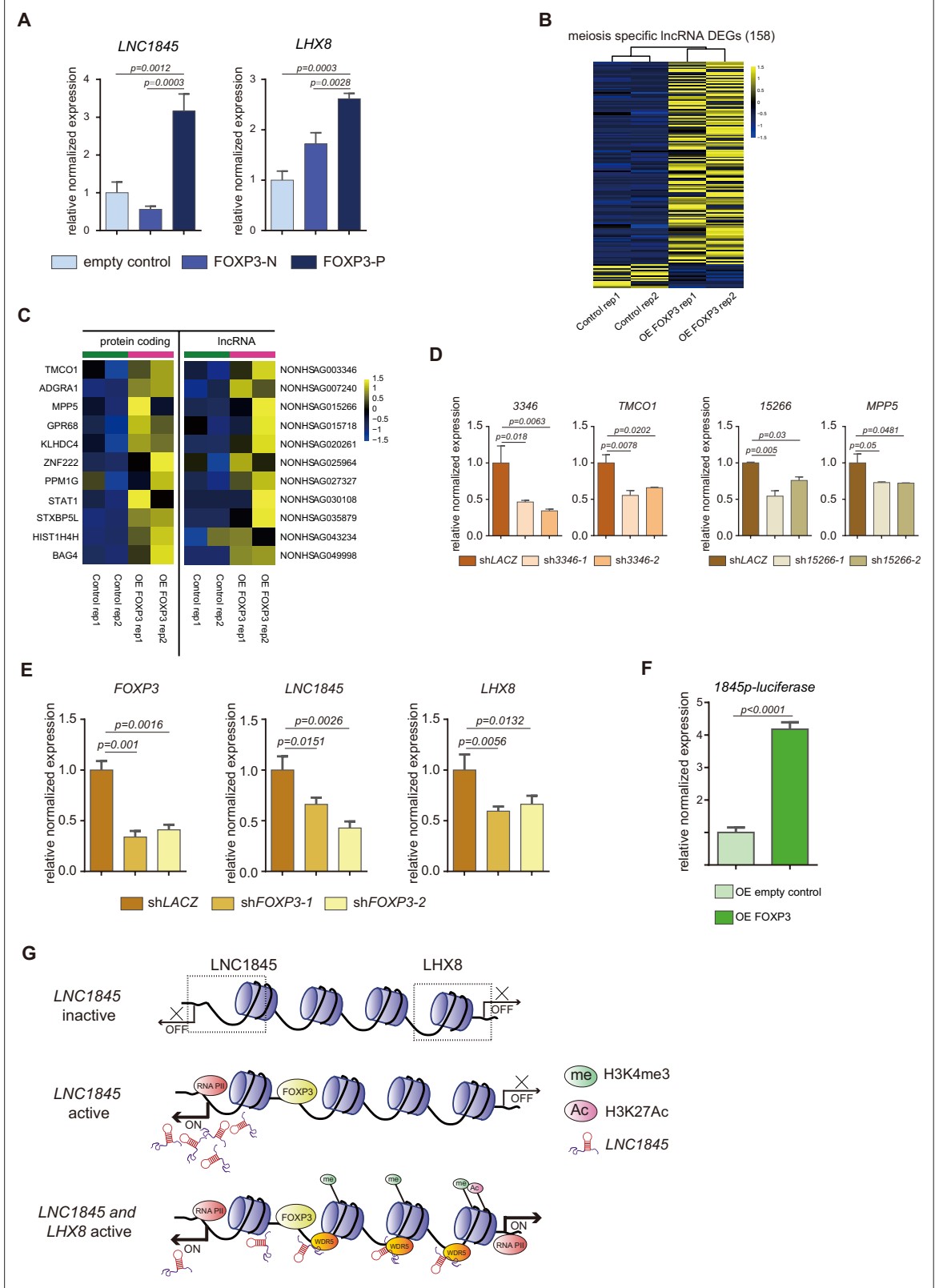

**Figure 7.** FOXP3 upregulates expression of *LNC1845* and other long non-coding RNAs (lncRNAs). (**A**) RT-qPCR analysis of *LNC1845* and *LHX8* in primordial germ cell-like cells (PGCLCs) overexpresses FOXP3. (**B**) Heatmap of differentially expressed ms-lncRNAs in control or OE FOXP3 groups, rep1 and rep2 represent two independent populations. (**C**) Heatmap of upregulated divergent lncRNA–mRNA pairs in OE FOXP3 groups, rep1and rep2 represent two independent populations. (**D**) Expression analysis by RT-qPCR of lncRNA and their divergent protein-coding genes in lncRNA knockdown

*Figure 7 continued on next page*

*Figure 7 continued*

cells (*n* = 3). (**E**) Expression analysis by RT-qPCR of *LNC1845* and *LHX8* in *FOXP3* knockdown PGCLCs (*n* = 3). (**F**) Luciferase assay of FOXP3 on *LNC1845* promotor activity. 293 FT cells were transiently transfected with empty control or FOXP3 expression vector in combination with luciferase vectors, *n* = 3. (**G**) The FOXP3-*LNC1845*-LHX8 regulatory model. The upstream transcription factors, including FOXP3, could upregulate *LNC1845,* and then *LNC1845* could induce H3K4me3 and H3K27Ac in the regions near the *LHX8* transcription start site, which in turn helps activate *LHX8* expression. In (**A**), (**D**), (**E**), and (**F**), the *y*-axis represents relative mean expression normalized to *GADPH* and control cells. Error bars indicate the mean ± standard deviation (SD), three independent experiments were carried out. One-way analysis of variance (ANOVA) was used in (**A**), (**D**), and (**E**), and Student's *t*-test was used in (**F**) for the comparisons.

The online version of this article includes the following figure supplement(s) for figure 7:

**Figure supplement 1.** FOXP3 upregulates expression of *LNC1845* and other long non-coding RNAs (lncRNAs).

Overexpressing FOXP3 obviously activated *LNC1845* promoter compared with overexpressing empty control group (***Figure 7F***). Taken together, FOXP3 was found to specifically express in germ cells rather than gonadal somatic cells in human fetal gonads, and overexpressing FOXP3 alone upregulates expression of many meiotic specific lncRNAs by direct promoter binding.

## Discussion

Recent studies have identified the expression of lncRNAs in *Drosophila* and mouse testes (***Wen et al., 2016***; ***Wichman et al., 2017***), but the functions and regulation mechanisms of these lncRNAs are not clear. Furthermore, many lncRNAs expressed in model organisms are not conserved in humans. Another recent study reported differential expression profiles of human MII oocytes and cumulus cells (***Bouckenheimer et al., 2018***), but the lncRNA expression profiles in other developmental stages were not included. In contrast, cell-type-specific and developmental-stage-specific lncRNA expression profiles during human gonadal development from 4 WG to 26 WG in both sexes are elucidated.

During human gonadal development, 20,323 lncRNAs in germ and somatic cells, which have not been annotated in GENCODE and RefSeq (NCBI), were identified. Although an increasing number of human lncRNAs have been annotated in several annotation databases such as NONCODE, many novel lncRNAs have not been fully identified and systematically annotated due to the lack of examination in specific tissues or developmental stages (***Uszczynska-Ratajczak et al., 2018***). Our study provides cell-type specific and developmental timing expression of lncRNAs for future studies of lncRNAs in human gonadal germ cells and somatic cells. Interestingly, the ratios of lncRNA versus protein-coding genes in fGCs and mGCs are higher than the ratios in the somatic cells, suggesting that lncRNAs may play more critical roles or have regulatory functions in germ cells than gonadal somatic cells during gonadal development. However, it is not clear whether all of these lncRNAs are essential for germ cell development. Further studies are needed to validate and elucidate the roles of these lncRNAs.

Cell-type classification is widely used in single-cell study to define and understand cellular state during dynamic developmental process (***Li et al., 2017***; ***Wang et al., 2020***). Current cell-type classification mostly relies on transcriptional expression profiles of coding genes. However, expression profiles of coding genes may not be different enough to distinguish two cell types which are closely similar. Integrating expression profiles of non-coding genes such as lncRNA will enable us to further dissect different cellular states in order to more accurately describe the changes happen during developmental programs such as germ cell development. A comprehensive analysis identifies a new population of cells after integrating lncRNA expression profile. The challenge is the sequencing depth of many single-cell study may not cover enough non-coding genes for this integration.

Another interesting feature of lncRNA profiles reveals that the majority of lncRNAs expressed in hGCs belong to divergent XH biotypes. Recent studies of divergent lncRNAs showed that many of them are involved in regulating the expression of neighboring genes (***Canzio et al., 2019***; ***Kimura et al., 2017***; ***Luo et al., 2016***; ***Yin et al., 2015***). A previous report showed that many lncRNAs have local regulation of nearby protein-coding genes (***Engreitz et al., 2016***), and divergent lncRNAs tend to co-express with developmental and transcription regulator genes (***Luo et al., 2016***). According to lncRNAs landscape, both fGCs and mGCs expressed more divergent lncRNAs than the other biotypes, suggesting that the divergent lncRNAs or the transcriptional activities of these lncRNA may affect the transcriptional expression of the neighboring coding genes. More examples of divergent lncRNAs serving as regulatory lncRNAs during hGC development are expected to be discovered in the future.

Focusing on one divergent lncRNA expressed in both in vivo and in vitro differentiated germ cells, the mechanism of how *LNC1845* regulates *LHX8* through *cis*-activation is elucidated. Our experimental results utilizing multiple approaches all suggest that *LNC1845* RNA regulates LHX8 expression through *cis* activation. Previous studies have shown that *Lhx8*-deficient mice exhibit a lack of transition from primordial to growing follicles (*Choi et al., 2008*; *Pangas et al., 2006*). *Lhx8*-null mouse ovaries misexpress many oocyte-specific genes, such as growth differentiation factor-9 (*Gdf9*), POU class 5 homeobox (*Pou5f1*), and newborn ovary homeobox (*Nobox*) (*Choi et al., 2008*). Several single-nucleotide polymorphisms have been identified in *LHX8* associated with primary ovarian insufficiency patients (*Bouilly et al., 2016*; *Qin et al., 2008*). One of the reported missense mutations (A325V) in *LHX8* caused lower transcriptional expression of LIN28A (*Bouilly et al., 2016*). A new regulatory lncRNA at the 5′UTR of *LHX8* acting through the *cis*-activation of *LHX8* transcription was identified. Based on this model, a loss-of-function mutation occurring in *LNC1845* might lower the expression of *LHX8* during human folliculogenesis and might lead to premature ovarian insufficiency.

The high number of lncRNA expression suggests that there may be a common regulator of transcription factors during hGC development, especially in late PGCs and after cell entering meiosis. Using PROMO prediction and in vitro validation, we confirmed that FOXP3 alone is able to transcriptionally activate many lncRNAs expressed in hGCs. FOXP3 was reported to play important roles during regulatory T-cell development in previous studies (*Gavin et al., 2007*; *Weigel et al., 1989*), and FOXP3 polymorphisms are associated with human female endometriosis and male infertility (*Brunkow et al., 2001*; *Piccirillo, 2020*). In our study, FOXP3 specifically expresses in germ cells rather than in gonadal somatic cells in human gonads. Moreover, overexpression FOXP3 alone upregulates expression of many meiotic specific lncRNAs. Although no study has reported a regulatory function of FOXP3 on human fGC development, previous studies reported knockout mice died 6–8 weeks after birth. It is possible that the effect on reproduction and germ cell development has not been carefully examined. Conditional Foxp3 knockout mice may be needed to further dissect its function in lncRNA regulation.

In summary, an unbiased bioinformatic analysis provides a new and comprehensive profiling of lncRNAs expressed during human gonadal development. Using in vitro differentiated germ cells, the regulatory role of *LNC1845* expressed in hGCs and its relationship with its neighboring gene, *LHX8*, were elucidated. The production of *LNC1845* changes the histone modification level of H3K4me3 and H3K27Ac and may prepare a more open chromatic structure for the transcriptional activation of the neighboring gene, *LHX8* (*Figure 7G*). *Cis*-activating lncRNAs have been discovered in some studies (*Gil and Ulitsky, 2020*), but this is the first example in hGCs.

The functional roles of lncRNAs in germ cell development and fertility have gained more attentions (*Tzur, 2022*). However, lncRNA is not the only non-coding RNAs important for germ cell developments, miRNAs have recently been elucidated to regulate proliferation of hPGCs (*Yan et al., 2022*). The regulation of transcriptional level is not always the most important for gene expression in germ cells, recent studies also underlying the importance of translational regulation in oocyte maturation and spermiogenesis (*Hu et al., 2022*; *Kang et al., 2022*). Hence, germ cells employ many unique posttranscriptional regulatory mechanisms to ensure proper gene expressions throughout the whole developmental processes.

## Materials and methods

All reagents and resources are listed on an appendix attached separately.

### hESC culture and hPGCLC induction

The hESC line H9 (female XX line) was purchased from WiCell, Inc Cells are propagated as described previously (*Jung et al., 2017*; *Kee et al., 2009*). In brief, undifferentiated hESCs were expanded on irradiated MEFs at 37°C with 5% $CO_2$ in hESC medium (KnockOut DMEM [Invitrogen, 10829018] with 20% knockout serum replacement [Invitrogen, 10828028] plus 1 mM L-glutamine, 0.1 mM nonessential amino acids, and 8 ng/ml recombinant human basic FGF [R&D systems]). H9 cells and 293 FT cells in this study were authenticated by STR profilings and were tested negative for mycoplasma contamination.

To differentiate hESCs into hPGCLCs, ~$3 \times 10^4$ undifferentiated hESCs (~30% confluency in a 6-well plate) were seeded onto one well of a 6-well plate coated with Matrigel. Adherent differentiation

began by treating the hESCs with differentiation medium (KODMEM supplemented with 10% FBS, 1 mM L-glutamine, 0.1 mM nonessential amino acids, 50 ng/ml BMP4 and BMP8a [R&D systems]) for 2 hr at 37 °C with 5% $CO_2$. This procedure was followed by transduction with lentiviral supernatant expressing DAZL-P2A-mCherry for 6 hr, then the differentiated hESCs were incubated with differentiation medium containing 50 ng/ml BMP4 and BMP8a for 12 days.

The cells were harvested by FACS for mCherry-positive cells, and mCherry-negative cells were also harvested as negative control group. The cells were digested with TrypLE Express (Invitrogen, 12605028). The single-cell suspension for FACS was prepared with FACS medium (10% FBS in DPBS) and filtrated through a BD cell sorter. Cell sorting was proceeded on a high-speed cell sorter (Influx, BD) and was sorted to collecting tube containing differentiation medium.

## Overexpression and silencing of genes by lentivirus in hESCs

Overexpression vectors of DAZL, *LNC1845*, TFAP2A, and FOXP3 were constructed by cloning the full length of gene-coding sequence, then inserted into the lentiviral vector p2k7 with E1Fα promoter as in our previous study (*Liang et al., 2019*). *LNC1845*, *LNC3346*, *LNC15266*, *FOXP3*, and *WDR5* targeting shRNA were designed by using Public TRC Portal (the RNAi Consortium, Broad Institute). Their expressions were driven by H1 promoter in a lentiviral vector containing a Ubc promoter-drived GFP fluorescent reporter. Knockdown or overexpression vector was packaged into lentivirus and infected into the cells. Sequences of shRNAs are listed in *Supplementary file 10*.

## CRISPR-mediated activation (CRISPR-on)

CRISPR-on was performed as previously described (*Luo et al., 2016*). Plasmids expressing dCas9-VP64 (Addgene #61425), MS2-P65-HSF1 (Addgene #61426), and sgRNA (fused with MS2) (Addgene #61427) were co-transfected into cells by VigoFect (Vigorous Biotechnology, T001). Sequences of gRNAs are listed in *Supplementary file 10*. Cells were collected 2 days after transfection followed by RNA isolation and RT-qPCR analysis.

## Immunofluorescence of cultured cells

hESCs (H9) were cultured on coverslip coated with 1% Matrigel or sorted by FACS enrichment and collected onto a slide by Cytospin (800 rpm for 5 min). After the hESCs were differentiated as described above, the adherent differentiated cells were rinsed twice in phosphate-buffered saline (PBS), pH 7.4, fixed with 4% paraformaldehyde for 15 min, and permeabilized with 0.3% Triton X-100 in PBS for 15 min at room temperature. Permeabilized cells were blocked in 10% donkey serum in PBS for 1 hr with gentle agitation. The blocked cells were then incubated overnight at 4°C with the primary antibody at the dilution listed in Key Resource Table. The coverslips were washed with PBS five times and then incubated with the secondary antibody listed in Key Resource Table for 1 hr at room temperature. Then, the coverslips were washed as described above, stained with 2 mg/ml DAPI(Diamidinyl phenylindole) for 10 min at room temperature, and then washed again as described above. The slides were mounted with ProLong Gold Antifade (Life Technologies) and observed under a confocal microscope.

## CRISPR/Cas9-mediated knock-out and knock-in at *LNC1845*

We analyzed the predicted LNC1845 sequences from NONCODE and GENCODE database (chr1:75,122,517–75,123,927), and our RNA-seq mapping track showed the mapping sequences covered chr1:75,122,766–75,123,909 (18 bp less than the database). Combining the database sequences and our RNA-seq results, we marked the 75,123,927 at chromosome 1 as the 5′ end of LNC1845. Plasmids expressing Cas9/sgRNAs and donor sequences were co-transfected into H9 by lipofectamine 3000. The ratio of Cas9/sgRNAs versus donor is 1:4. For *LNC1845* knockout, we designed a donor plasmid including two homology arms and two loxp locus flanking the sequence encoding puromycin drug resistance, replacing LNC1845 transcript but keeping the 5′ upstream and 3′ downstream of LNC1845 (*Figure 5A*). After picking out single colony and analysis, two homozygous knock-out ESC lines were treated with Cre to excise puromycin-resistant cassette.

For knock-in of a 3×polyA transcription stop signal, the targeting vector contains two homology arms and an expression cassette of puromycin-resistant gene flanked by two loxp sites, followed by 2×SV40 polyA signal sequence and a BGH polyA signal. This plasmid was constructed by editing

3×pA KI donor plasmid from Prof. Shen laboratory (*Yin et al., 2015*). It was co-transfected with Cas9/sgRNAs which targets the immediate downstream of *LNC1845* TSS. After picking out single colony and analysis, two homozygous knock-in ESC lines were treated with Cre to excise puromycin-resistant cassette.

For EF1α promoter knock-in, the targeting vector contains 5′ and 3′ homology arms (849 and 620 bp in length respective) upstream of puromycin resistance gene and downstream of EF1α promoter. The targeting vector was co-transfected with Cas9 and sgRNAs which targets immediate upstream of *LNC1845*. Out of 18 clones analyzed, two heterozygous knock-in hESCs were isolated. Primers and gRNA sequences are listed in *Supplementary file 10*.

## Western blot analysis

The procedure for the western blot analysis was previously described (*Jung et al., 2017*), and the specific antibodies are listed in Key Resource Table. In brief, the differentiated cells were washed with 2-ml cold DPBS (Corning, R21-031-CV), then scraped from the plate in 1-ml DPBS plus 2×protease inhibitors (Complete Mini, Roche) and transferred immediately to a 1.5-ml tube on ice. The cell suspension was then spun at 3500 rpm in a microcentrifuge for 5 min and the supernatant was discarded. The cell pellet was resuspended with 100-ml RIPA buffer (50 mM Tris, 150 mM NaCl, 0.5% sodium deoxycholate, 1% NP-40, 0.1% sodium dodecyl sulfate [SDS], pH 8) plus 2×protease inhibitors (Complete Mini, Roche). The cell pellet suspension was pipetted rigorously at least 10 times, then vortexed for 30 s. The suspension was again spun down for 10 min at the 10,000 rpm. The supernatant was measured for protein concentration and denatured in Laemmli buffer at 95°C for 5 min, then loaded onto a 10% SDS–polyacrylamide gel electrophoresis (PAGE gel). The SDS–PAGE gels were run at 80 volts for 30 min and 120 V for 1 hr, and transferred to a PVDF(Polyvinylidene Fluoride) membrane for 1 hr at 100 V in transfer buffer. Transferred blots were blocked in 5% non-fat milk for 1 hr at room temperature. The blot was subjected to primary antibody incubation overnight at 4°C, followed by two quick rinses and five washes for 5 min in TBST (TBS, pH 7.4 with 0.1% Tween-20). The blot was subjected to secondary antibody incubation for 1 hr at room temperature. Immobilon Western Chemilum HRP Substrate (Merck Millipore, WBKLS0100) was used to detect the HRP signal and the western blot image was collected using Chemidoc XRS (Bio-Rad).

## RNA extraction, reverse transcription, and quantitative real-time PCR

Total RNA was extracted using the TRIzol Reagent (Life Technologies) and the cDNA was synthesized using the PrimeScript RT reagent Kit with gDNA Eraser (TaKaRa) according to the manufacturer's protocols. Quantitative PCR was performed using the TransStart Top Green qPCR SuperMix (TransGen) on a CFX96 Real-time PCR machine (Bio-Rad). The gene expression level was first normalized to the housekeeping gene glyceraldehyde 3-phosphate dehydrogenase and then normalized to the expression of a control sample in each set of experiments using Bio-Rad CFX Manager program and relative expression formulation d*C*(*t*). Statistical analysis was performed using GraphPad Prism 6. Real-time PCR primers for all gene sequences are listed in *Supplementary file 10*.

## RNA fluorescent in situ hybridization

RNA FISH was performed using Ribo Fluorescent In Situ Hybridization Kit (RIBOBIO, C10910). Cells were rinsed briefly in 1 ml DPBS and then fixed in 4% paraformaldehyde for 10 min at room temperature. Cells were permeabilized in DPBS containing 0.3% Triton X-100 for 15 min at 4°C, then washed in DPBS for 5 min. 200 ml of Pre-hybridization Buffer was added at 37°C for 2 hr. Hybridization was carried out with a FISH probe at 42°C in the dark for 1 hr. The cells were washed at 50°C for three times with Wash Buffer I (4× SSC(Saline Sodium Citrate buffer) with 0.1% Tween-20), once each with Wash Buffer II (2× SSC), Wash Buffer III (1× SSC) in the dark for 5 min and once with 1× DPBS at room temperature. Then the cells were stained with DAPI in the dark for 10 min. *LNC1845* FISH probes were designed and synthesized by RiboBio Co, Ltd and the probe number is lnc1100816. All images were obtained with a fluorescence or confocal microscope (Nikon).

## In vitro transcription and RNA transfection

RNA in vitro transcription was performed using Ribo RNAmax-T7 kit (RIBOBIO, R11073). The reaction mixture was prepared with linearized template DNA and T7 Enzyme Mix, incubated at 37°C for 2 hr,

and DNase I was added to remove template DNA. The RNA product was further purified and diluted to 1 μg/μl for transfection.

RNA transfection was performed using Entranster-R4000 (4000-3) according to the manufacturer's protocols.

## In vitro transcription and translation

In vitro transcription and translation were performed using TnT Quick Coupled Transcription/Translation Systems (L1170) coupled with FluoroTect GreenLys in vitro Translation Labeling System (L5001). In brief, the reaction components including DNA templates, TnT Quick Master Mix, and FluoroTect GreenLys tRNA were assembled in a 1.5-ml microcentrifuge tube and incubated at 30°C for 90 min. The results were analyzed by gel analysis of translation products with Typhoon 8600 fluorescent imaging instrument.

## 3′ RACE

The RNA-ligase-mediated RACE (RLM-RACE) was carried out with total RNA extracted from PGCLCs and was used to determine the transcription stop points and the size of the *LNC1845* transcripts. Rapid amplification of 3′ cDNA ends was carried out using a FirstChoice RLM-RACE kit (Ambion), according to the manufacturer's instructions. RACE was performed by using a gene-specific primer. When no distinct fragment sizes were observed, the PCR product was diluted and amplified with a nested gene-specific primer, and single PCR bands were directly sequenced. Primer sequences are listed in *Supplementary file 10*.

## ULI-NChIP

ULI-NChIP was performed for each histone modification mark according to the published method (*Brind'Amour et al., 2015*). Briefly, 100,000 cells were FACS-sorted, spun down, and resuspended in nuclear isolation buffer. Chromatin were digested using MNase and immunoprecipitated by incubating with antibody against H3K4me3 (Millipore, 04-745) and H3K27ac (Cell Signaling Technology, 8173) attached protein A:protein G Dynabeads. Next, chromatin were eluted and purified using PCI (phenol:chloroform:isoamyl alcohol). Finally, raw ChIP material was re-purified with 1.8× volume of AMpure XP beads before library construction.

## Library construction

For ULI-NChIP-seq, 5 ng or 85% of the raw ChIP material was used for library construction using the NEBNext DNA Library Prep Kit and NEBNext Multiplex Oligos for Illumina. In brief, samples were end-repaired, 5′ phosphorylated and dA-tailed, then ligated with NEBNext adaptor. Ligated fragments were amplified using indexed primers (Illumina) and size selected with Ampure XP DNA purification beads.

## ChIP-seq data processing and analysis

In brief, raw paired-end sequencing reads were assessed the sequencing quality using FastQC, then the adaptors were trimmed and removed low-quality reads. Next, the cleaned reads were mapped to the Gencode.v29 of the human reference genome using Hisat2 with default parameters. Next, samtools was used to remove PCR duplicates and we represented each histone modification mark as continuous enrichment values of 100 bp bins across the genome using bamCoverage in Deeptools2. The enrichment of each histone modification mark was defined as the RPKM and then viewed in IGV software. We binned ChIP and Input into the same bins (10 bp size), then normalized the reads overlapping the bins by RPKM for both ChIP and Input to account for differences in sequencing depth. After normalizing, we used the log2 ratio between the bins as a measurement of enrichment over input.

## Subcellular fractionation

Cells were washed with DPBS(Duchenne phosphate buffer solution) and incubate in the lysis buffer (150 mM NaCl, 0.1% NP-40, 0.5% sodium deoxycholate, 0.1% SDS, 50 mM Tris, 1 mM DTT(dithiothreitol), 5 mM $Na_3VO_4$, 1 mM phenylmethylsulfonyl fluoride, 10 μg/ml trypsin inhibitor, 10 μg/ml aprotinin, 5 μg/ml leupeptin; pH 7.4). The lysate was centrifuged at 12,000 × *g* for 30 min at 4°C and the

supernatant was collected as whole-cell lysate. The cytosolic fractions were prepared using the Mitochondria/Cytosol Fractionation Kit (BioVision). In brief, cells were centrifuged at 600 × *g* for 5 min at 4°C, resuspended in the extraction buffer containing DTT and protease inhibitors, incubated on ice for 10 min and then homogenized with an ice-cold Dounce tissue grinder. Unbroken cells and nuclei were pelleted by centrifugation at 700 × *g* for 10 min at 4°C. The supernatant was centrifuged at 10,000 × *g* for 30 min to yield a cytosol-enriched fraction. The nucleic fractions were prepared using Nuclear and Cytoplasmic Protein Extraction kit (Beyotime Biotech). Cells were mixed with the cytoplasmic extraction buffer on ice for 15 min. Suspension was shaken vigorously and centrifuged at 12,000 × *g* for 5 min at 4°C. The supernatant was discarded and the pellet was resuspended in nuclear extraction buffer on ice for 30 min. The resulting supernatant was used as nucleic fractions following centrifuge at 12,000 × *g* for 10 min. All subcellular fractions were stored at −80°C.

## Quantification and statistical analysis

Statistical details of analysis including statistical test used, value of *n* and statistical significance were all described in the figure legends.

### Analyzing single-cell RNA-seq data

Raw single-cell RNA-seq data were downloaded from GEO with accession GSE86146 and processed according to Ji Dong's methods. Briefly, sequencing reads were firstly separated by specific cell barcode and removed the template switch oligo sequence and polyA tail. Next, the clean reads were mapped to the Gencode.v29 of the human reference genome using Tophat with the GTF file containing both the annotations of protein-coding genes from Gencode.v29 and lncRNA genes from NONCODEv5. After mapping reads deduplication, the read counts of each lncRNA are calculated using HTSeq package with distinct UMI information and then transformed into TPM. Finally, we extracted the lncRNA transcriptome of 1845 single cells that are identified and clustered by Tang group (*Li et al., 2017*) and classified them into each cell type. A lncRNA or a protein-coding gene is defined as expressed if it is expressed in more than two cells and its maximum TPM >3. The classification of annotated and unannotated lncRNA genes are performed by comparing the lncRNA genes genomic location between the NONCODEv5_hg38.lncAndGene.bed (from NONCODE database) and gencode.v29.long_noncoding_RNAs.gtf (from Gencode database). A lncRNA is defined as annotated if it is included in these two databases, and as unannotated if it is only recorded in NONCODE database. The distribution of lncRNA genes is analyzed by using annotatePeaks.pl with hg38.basic. annotation in HOMER software. Comparison of weekly expressed lncRNA and protein-coding genes number between germ cells and gonadal somatic cells are plotted using R package ggplot2 and the statistical significance is evaluated using Student's *t*-test. Cell-type-specific lncRNA genes are defined as they expressed in no more than one cell of other cell types and viewed their expression in Treeview. Cell-type-specific lncRNA genes are classified into six subtypes according to their transcription orientation and genomic locations relative to the nearest protein-coding genes, and the lncRNA–mRNA pairs expression Pearson correlation coefficient is calculated using cor() function of R. The cells used for Pearson correlation analysis are selected under the criteria that they must expressed the specific lncRNA of the lncRNA–mRNA pairs.

### In vitro bulk RNA-seq library preparation, sequencing, and data processing

After RNA extraction, we used the VAHTS mRNA-seq v2 Library Prep Kit (Vazyme NR611-02) for library construction. Briefly, after polyA selection the RNA went through reverse transcription and the first-strand cDNA synthesis. Then the cDNA is amplified using a limited number of PCR cycles and Tn5 tagmentation is used to construct sequencing libraries. After PCR amplification the libraries were size-selected and sequenced using 150-bp paired-end sequencing on an Illumina NovaSeq 6000 platform.

Raw sequencing reads were cleaned after removing low-quality reads and trimmed adaptors. The quality of cleaned reads was assessed using FastQC and MultiQC. Next, clean reads were mapped to the Gencode.v29 of the human reference genome using Hisat2 with the GTF file contains both the annotations of protein-coding genes from Gencode.v29 and lncRNA genes from NONCODEv5. Read counts per gene were obtained from the aligned reads using htseq-count of HTSeq package. DESeq2 was used for normalization and differential gene expression analysis.

We normalized read counts into FPKM (Fragments Per Kilobase per Million) for the hierarchical clustering of differentially expressed genes and heatmaps were generated using pheatmap package with row z-score normalization.

## Nonlinear dimensional reduction (t-SNE)

The R package called Seurat was applied to perform nonlinear dimensional reduction(t-SNE) on the coding transcriptome only and combined transcriptome data, respectively, using normalized TPM expression values. Top 2000 high variable genes/lncRNAs were used for the principle component analysis. Jackstraw with 500 replicates analysis was used to select principal components (PCs) to separate the cells. We select PCs 1–10 to perform the RunTSNE function and the FindClusters function (resolution = 0.6), obtaining clusters 0–7 and 0–8, respectively. Seurat FindAllMarkers function based on normalized TPM expression values was used to identify unique cluster-specific marker genes and lncRNAs.

## Pseudotime analysis

For germ cells, pseudotime was generated by Monocle package (v2.10.1) based on top 2000 high variable genes/lncRNAs identified by Seurat following the default settings. Single-cell pseudotime trajectories were constructed with the Monocle2 package (v2.10.1) according to the operation manual (*Qiu et al., 2017*). Briefly, the UMI count matrices of the germ cells were input as the expr_matrix, and meta_data was input as the sample sheet. Then, highly variable protein-coding genes and lncRNAs identified by Seurat were chosen to define a cell's progress. In this step, ordering genes that were expressed in less than 5 cells and had a p value bigger than 0.001 were excluded. DDRTree was used to reduce the space down to one with two dimensions, and all cells were ordered with the orderCells function.

## GO analysis

GO analysis was performed using the DAVID Functional Annotation Bioinformatics Microarray Analysis tool (http://david.abcc.ncifcr.gov/; *Huang et al., 2009*).

## Code availability

Custom scripts used in this study are publicly available at https://github.com/Maggie159123/elife_2022, (*Wang, 2022* copy archived at swh:1:rev:3e9179321fe4adaa1d7cb7e3fccafd011cff381e).

## Acknowledgements

We thank Prof. Xiaohua Shen at Tsinghua University for giving us plasmids and technical supports. Research funding was provided by the Ministry of Science and Technology of China (2021YFA0719301, 2018YFA0107703); the National Natural Science Foundation of China (82071597); Tsinghua-Peking Center for Life Sciences; and the Cross-Discipline Foundation of Tsinghua University.

## Additional information

### Funding

| Funder | Grant reference number | Author |
| --- | --- | --- |
| Ministry of Science and Technology of the People's Republic of China | 2021YFA0719301 | Kehkooi Kee |
| National Natural Science Foundation of China | 82071597 | Kehkooi Kee |
| Ministry of Science and Technology of the People's Republic of China | 2018YFA0107703 | Kehkooi Kee |

| Funder | Grant reference number | Author |
|--------|------------------------|--------|

The funders had no role in study design, data collection, and interpretation, or the decision to submit the work for publication.

## Author contributions

Nan Wang, Conceptualization, Data curation, Formal analysis, Validation, Investigation, Visualization, Writing – original draft; Jing He, Data curation, Formal analysis, Validation, Investigation, Methodology, Writing – original draft; Xiaoyu Feng, Formal analysis, Investigation, Visualization, Methodology; Shengyou Liao, Yi Zhao, Data curation, Formal analysis, Supervision, Methodology; Fuchou Tang, Data curation, Formal analysis, Writing – review and editing; Kehkooi Kee, Conceptualization, Resources, Supervision, Funding acquisition, Validation, Writing – original draft, Project administration, Writing – review and editing

## Author ORCIDs

Kehkooi Kee ⓘ http://orcid.org/0000-0001-6926-7203

## Decision letter and Author response

Decision letter https://doi.org/10.7554/eLife.78421.sa1
Author response https://doi.org/10.7554/eLife.78421.sa2

## Additional files

### Supplementary files

• Supplementary file 1. Weekly expressed lncRNAs of female and male germ cells and gonadal somatic cells.

• Supplementary file 2. Human germ cells and gonadal somatic cells specific lncRNAs.

• Supplementary file 3. Human germ cell each stage-specific lncRNAs.

• Supplementary file 4. Feature protein-coding genes or lncRNAs of each cluster.

• Supplementary file 5. Female germ cell (fGC) meiotic and male germ cell (mGC) mitotic arrest lncRNA–mRNA pairs and expression correlation.

• Supplementary file 6. Upregulated lncRNA genes in primordial germ cell-like cells (PGCLCs).

• Supplementary file 7. Upregulated protein-coding genes in primordial germ cell-like cells (PGCLCs).

• Supplementary file 8. Differentially expressed protein-coding genes in LNC1845 KO primordial germ cell-like cells (PGCLCs).

• Supplementary file 9. Differentially expressed protein-coding genes in LHX8 OE primordial germ cell-like cells (PGCLCs).

• Supplementary file 10. Oligonucleotides.

### Data availability

Raw single-cell RNA-seq data are available in GEO with accession GSE86146.

The following previously published dataset was used:

| Author(s) | Year | Dataset title | Dataset URL | Database and Identifier |
|-----------|------|---------------|-------------|-------------------------|
| Li L, Dong J, Yan L, Qiao J, Tang F | 2017 | Single-Cell RNA-Seq Analysis Maps Development of Human Germline Cells and Gonadal Niche Interactions | https://www.ncbi.nlm.nih.gov/geo/query/acc.cgi?acc=GSE86146 | NCBI Gene Expression Omnibus, GSE86146 |

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

# Appendix 1

## Appendix 1—key resources table

| Reagent type (species) or resource | Designation | Source or reference | Identifiers | Additional information |
|---|---|---|---|---|
| Gene (*Homo sapiens*) | *LIM homebox 8(LHX8)* | Human genome informatics | Ensembl:ENSG 00000162624 | |
| Gene (*Homo sapiens*) | *LNC1845(AC099786.3)* | Human genome informatics | NONHSAG001845. 2/ENSG00000261213.1 | |
| Cell line (*Homo sapiens*) | H9 LNC1845 KO stem cells | This paper | | LNC1845 KO cell line generated by CRISPR/Cas9 |
| Cell line (*Homo sapiens*) | H9 LNC1845 polyA KI stem cells | This paper | | LNC1845 polyA KI cell line generated by CRISPR/Cas9 |
| Cell line (*Homo sapiens*) | H9 LNC1845 EF1α KI stem cells | This paper | | LNC1845 EF1α KI cell line generated by CRISPR/Cas9 |
| Cell line (*Homo sapiens*) | PA-1 cells | This paper | | Human ovary teratocarcinoma Cell Line |
| Cell line (*Homo sapiens*) | 293 FT cells | Thermo Fisher Scientific | Cat# R70007 | |
| Cell line (*Mus musculus*) | Mouse embryonic fibroblasts | This paper | | Mouse embryonic fibroblasts isolated from embryonic day 13.5 |
| Recombinant DNA reagent | P2k7-EF1α-DAZL2-P2A-mCherry | This study | | Methods |
| Recombinant DNA reagent | P2k7-EF1α-mCherry | This study | | Methods |
| Recombinant DNA reagent | P2k7-EF1α-1845 | This study | | Methods |
| Recombinant DNA reagent | P2k7-EF1α-FOXP3 | This study | | Methods |
| Recombinant DNA reagent | P2k7-EF1α-TFAP2A | This study | | Methods |
| Recombinant DNA reagent | P2k7-EF1α-FOXP3-P2A-mCherry | This study | | Methods |
| Recombinant DNA reagent | P2k7-EF1α-FOXP3-3xFLAG-P2A-mCherry | This study | | Methods |
| Recombinant DNA reagent | pEASY-T7 promoter | This study | | Methods |
| Recombinant DNA reagent | pEASY-T7-1845 | This study | | Methods |
| Recombinant DNA reagent | pX335-U6-Chimeric_BB-CBh-hSpCas9n(D10A) | Addgene | # 42335 | |
| Recombinant DNA reagent | PGK-puro-1845KO donor | This study | | Methods |
| Recombinant DNA reagent | SV40-puro-3xpolyA 1845 KI donor | This study | | Methods |
| Recombinant DNA reagent | SV40-puro-EF1α 1845 KI donor | This study | | Methods |
| Recombinant DNA reagent | H1-shLACZ-Ubc-GFP | This study | | Methods |
| Recombinant DNA reagent | H1-sh1845①-Ubc-GFP | This study | | Methods |
| Recombinant DNA reagent | H1-sh1845②-Ubc-GFP | This study | | Methods |
| Recombinant DNA reagent | H1-sh15266①-Ubc-GFP | This study | | Methods |
| Recombinant DNA reagent | H1-sh15266②-Ubc-GFP | This study | | Methods |

*Appendix 1 Continued on next page*

*Appendix 1 Continued*

| Reagent type (species) or resource | Designation | Source or reference | Identifiers | Additional information |
|---|---|---|---|---|
| Recombinant DNA reagent | H1-sh3346①-Ubc-GFP | This study | | Methods |
| Recombinant DNA reagent | H1-sh3346②-Ubc-GFP | This study | | Methods |
| Recombinant DNA reagent | H1-shFOXP3①-Ubc-GFP | This study | | Methods |
| Recombinant DNA reagent | H1-shFOXP3②-Ubc-GFP | This study | | Methods |
| Recombinant DNA reagent | H1-shWDR5①-Ubc-GFP | This study | | Methods |
| Recombinant DNA reagent | H1-shWDR5②-Ubc-GFP | This study | | Methods |
| Recombinant DNA reagent | H1-shWDR5③-Ubc-GFP | This study | | Methods |
| Recombinant DNA reagent | Lenti-dCAS-VP64-Blast | Addgene | # 61425 | |
| Recombinant DNA reagent | Lenti-sgRNA(MS2)-zeo backbone | Addgene | # 61427 | |
| Recombinant DNA reagent | Lenti MS2-P65-HSF1_Hygro | Addgene | # 61426 | |
| Recombinant DNA reagent | Lenti-sgRNA(MS2)–1845 gRNA 1 | This study | | Methods |
| Recombinant DNA reagent | Lenti-sgRNA(MS2)–1845 gRNA 2 | This study | | Methods |
| Recombinant DNA reagent | Lenti-sgRNA(MS2)–1845 gRNA 3 | This study | | Methods |
| Biological sample (*Homo sapiens*) | Human 17 w ovary RNA | BIOPIKE | | |
| Antibody | anti-DAZL (Mouse monoclonal) | Bio-Rad | Cat# MCA2336, RRID:AB_2292585 | (IF 1:50) |
| Antibody | anti-LHX8 (Rabbit polyclonal) | abcam | Cat# ab221882 | (IF 1:50; WB 1:1000) |
| Antibody | anti-WDR5 (Rabbit monoclonal) | Cell Signaling Technology | Cat# 13105S, RRID:AB_2620133 | (IF 1:200; WB 1:2000) |
| Antibody | anti-GAPDH (Mouse monoclonal) | Cwbio | Cat# CW0100M, RRID:AB_2801390 | (WB 1:5000) |
| Antibody | anti-FLAG (Rabbit monoclonal) | Sigma | Cat# F3165, RRID:AB_259529 | (IP 3 μg) |
| Antibody | anti-H3K27Ac (Rabbit monoclonal) | Cell Signaling Technology | Cat# 8173, RRID:AB_10949503 | (ChIP 1 μg/rxn) |
| Antibody | anti-H3K4me3 (Rabbit monoclonal) | Millipore | Cat# 04-745, RRID:AB_1163444 | (ChIP 1 μg/rxn) |
| Antibody | Goat anti-Rabbit IgG (H + L) Cross-Adsorbed Secondary Antibody, Alexa Fluor 488 (Goat polyclonal) | Thermo Fisher Scientific | Cat# A-11008, RRID:AB_143165 | (WB 1:5000) |
| Antibody | Goat anti-Mouse IgG (H + L) Highly Cross-Adsorbed Secondary Antibody, Alexa Fluor Plus 594 (Goat polyclonal) | Thermo Fisher Scientific | Cat# A32742, RRID:AB_2762825 | (WB 1:5000) |
| Antibody | Horseradish-labeled goat anti-rabbit IgG (H + L) (Goat polyclonal) | ZSGB-Bio | Cat# ZB-2301, RRID:AB_2747412 | (WB 1:5000) |

*Appendix 1 Continued on next page*

*Appendix 1 Continued*

| Reagent type (species) or resource | Designation | Source or reference | Identifiers | Additional information |
|---|---|---|---|---|
| Antibody | Peroxidase-Conjugated Goat anti-Mouse IgG (H + L) (Goat polyclonal) | ZSGB-Bio | Cat# ZB-2305, RRID:AB_2747415 | (WB 1:5000) |
| Chemical compound, drug | Puromycin | Sigma | Cat# P9620 | |
| Chemical compound, drug | Geneticin | Thermo Fisher Scientific | Cat# 10131035 | |
| Chemical compound, drug | Blasticidin | Thermo Fisher Scientific | Cat# 461120 | |
| Chemical compound, drug | ROCK inhibitor | stemRD | Cat# Y-005 | |
| Peptide, recombinant protein | Recombinant human BMP-4 | R&D | Cat# 314 BP | |
| Peptide, recombinant protein | Recombinant human BMP-8a | R&D | Cat# 1073-BPC | |
| Peptide, recombinant protein | Recombinant human FGF basic | R&D | Cat# 233-FB-001MG/CF | |
| Commercial assay or kit | Gateway LR Clonase II Kit | Thermo Fisher Scientific | Cat# 11791100 | |
| Commercial assay or kit | PrimeScript RT reagent Kit with gDNA Eraser | TaKaRa | Cat# RR047A | |
| Commercial assay or kit | Fluorescent In Situ Hybridization Kit | RIBOBIO | Cat# C10910 | |
| Commercial assay or kit | RNAmax-T7 kit | RIBOBIO | Cat# R11073 | |
| Commercial assay or kit | TnT Quick Coupled Transcription/ Translation Systems | Promega | Cat# L1170 | |
| Commercial assay or kit | FirstChoice RLM-RACE kit | Ambion | Cat# AM1700 | |
| Commercial assay or kit | AMPure XP beads | Beckman Coulter | A63881 | |
| Commercial assay or kit | NEBNext DNA Library Prep Kit | NEB | #E7645S | |
| Commercial assay or kit | NEBNext Multiplex Oligos | NEB | #E7335S | |
| Commercial assay or kit | VAHTS mRNA-seq v2 Library Prep Kit | Vazyme | NR611-02 | |
| Software, algorithm | GraphPad Prism 6.0 | GraphPad Prism | https://www.graphpad.com/ | |
| Software, algorithm | FlowJo | FlowJo | https://www.flowjo.com/ | |
| Software, algorithm | DAVID Bioinformatics Resources | *Huang et al., 2009* | https://david.ncifcrf.gov | |
| Software, algorithm | Coding Potential Calculator | *Kong et al., 2007* | http://cpc.cbi.pku.edu.cn/ | |
| Software, algorithm | Coding Potential Assessment Tool | *Wang et al., 2013* | http://rna-cpat. sourceforge.net/ | |
| Software, algorithm | PROMO | *Messeguer et al., 2002* | http://alggen.lsi.upc.es/ | |

*Appendix 1 Continued on next page*

*Appendix 1 Continued*

| Reagent type (species) or resource | Designation | Source or reference | Identifiers | Additional information |
|---|---|---|---|---|
| Software, algorithm | TopHat (v2.1.1) | *Trapnell et al., 2009* | http://ccb.jhu.edu/software/tophat/index.shtml | |
| Software, algorithm | HTSeq package | *Anders et al., 2015* | https://htseq.readthedocs.io/en/release_0.9.1/ | |
| Software, algorithm | GO (DAVID) | *Huang et al., 2009* | https://david.ncifcrf.gov/home.jsp | |
| Software, algorithm | HISAT2 (v2.1.0) | *Kim et al., 2015* | http://ccb.jhu.edu/software/hisat2/manual.shtml | |
| Software, algorithm | NONCODE (v5) | *Fang et al., 2017* | http://www.noncode.org/index.php | |
| Software, algorithm | Gencode (v29) | *Uszczynska-Ratajczak et al., 2018* | https://www.gencodegenes.org/ | |
| Software, algorithm | Samtools (1.3.1) | *Li et al., 2009* | http://samtools.sourceforge.net/ | |
| Software, algorithm | MACS2 | *Zhang et al., 2008* | https://pypi.org/project/MACS2/2.1.1.20160309/ | |
| Software, algorithm | IGV | *Thorvaldsdóttir et al., 2013* | http://www.igv.org/ | |
| Software, algorithm | Treeview | *Saldanha, 2004* | http://jtreeview.sourceforge.net/ | |
| Software, algorithm | R (3.5.1) | N/A | https://www.r-project.org/ | |
| Software, algorithm | pheatmap | N/A | R software | |
| Software, algorithm | DESeq2 | *Love et al., 2014* | R software | |
| Software, algorithm | ggplot2 | *Wickham, 2016* | R software | |
| Software, algorithm | Seurat (3.2.3) | *Satija et al., 2015* | R software | |
| Software, algorithm | Monocle (2.10.1) | *Qiu et al., 2017* | R software | |
| Other | Matrigel | Corning | Cat# 354277 | Matrigel Basement Membrane Matrix |
| Other | DAPI | Invitrogen | Cat# D1306 | Blue-fluorescent DNA stain |
| Other | TrypLE Express | Invitrogen | Cat# 12605010 | Trypsin substitution |
| Other | Dynabeads Protein A | Invitrogen | Cat# 10002D | Dynabeads Protein A for Immunoprecipitation |
| Other | Dynabeads Protein G | Invitrogen | Cat# 10003D | Dynabeads Protein G for Immunoprecipitation |
| Other | ProLong Diamond Antifade Mountant | Invitrogen | Cat# P10144 | Antifade Solution |
| Other | TRIzol Reagent | Invitrogen | Cat# 15596026 | RNA extraction reagents |
| Other | TransStart Top Green qPCR Mix | TransGen | Cat# AQ131-02 | qPCR reagents |
| Other | Lipofectamine 2000 | Invitrogen | Cat# 11668019 | Transfection reagents |
| Other | Lipofectamine 3000 | Invitrogen | Cat# L3000001 | Transfection reagents |

