## [Editor Report]

This manuscript provides a comprehensive analysis of expression patterns and genomic features of long non-coding RNAs (lncRNAs) in the human developing gonad. Using multiple genetic and molecular biology strategies in an in vitro system of female germ cell differentiation, the study further provides compelling evidence of a positive regulatory function of the LNC1845 lncRNA on its protein-coding neighbor LHX8, known to have a role in ovarian follicle development. This study has important interest to reproductive biologists and to the non-coding RNA community.

---

## [Decision Letter]

**Decision letter after peer review:**

Thank you for submitting your article "Single-cell profiling of lncRNAs in human germ cells and molecular analysis reveals transcriptional regulation of LNC1845 on LHX8" for consideration by *eLife*. Your article has been reviewed by 2 peer reviewers, and the evaluation has been overseen by a Reviewing Editor and Marianne Bronner as the Senior Editor. The following individual involved in the review of your submission has agreed to reveal their identity: Claire Rougeulle (Reviewer #2).

Essential revisions:

The two reviewers appreciated that the manuscript provides a very useful resource of human germ cell-specific lncRNAs for the community, and commended the authors for the impressive set of orthogonal approaches (KO, interrupted transcription, overexpression) they relied on to address the functional interactions between LNC1845 and its nearby gene LHX8. However, clarifications are needed to better support the claims, notably regarding the bioinformatic methodologies and the experimental strategies used for the functional genetic part.

Three main points, outlined below, have been identified as requiring particular attention. Please address them in a revised version of your manuscript. In addition, answer the individual comments of the reviewers in a detailed rebuttal letter, and provide the text clarifications and additional figures they request to include in the manuscript.

1. More insights into the cellular/phenotypic consequences of altered LHX8 expression is requested by the two reviewers, regarding the mention that LHX8 has a crucial function in ovarian development. One suggestion was to exploit the RNA-seq data to infer differentiation states of lnc1845 KO late PGCLCs. It seems also reasonable to ask the reviewers to provide information regarding the comparative efficiency of late PGCLC differentiation in presence or upon LHX8 downregulation (LNC1845 KO). The authors are welcome to provide other experimental means by which they could address the question of the functional consequences of the loss of lnc1845 and LHX8 on female PGCLC differentiation.

2. More information should be given as to how the 5' end of LNC1845 and proximal regulatory regions were defined (TF binding sites? 5'RACE?). The experimental strategy for editing and for overexpression should also be better described. This information is critical to support the functional analyses by deletion and by CRISPR ON.

3. The description of the datasets (developmental stages), methods for annotation of unknown lncRNAs, for quantification and for statistical analyses needs improvement.

*Reviewer #1 (Recommendations for the authors):*

My specific comments for each part of the study are summarized below:

1. Identification of lncRNAs expressed in human gonads

1.1 The first part of the manuscript annotating and cataloging lncRNAs in human germ cells requires some clarifications in respect of the number of identified lncRNAs. First, the authors should comment on the number of protein-coding genes in their data sets. Given that the human genome has in total 20.000-25.000 protein-coding genes, is it not surprising that about 14.000-16.000 protein-coding genes have been found to be expressed in gonads? Second, the same is valid for the number of lncRNAs that the authors identify to be expressed in these cells. Could it be that the high number of protein-coding and noncoding transcripts are due to a low expression threshold used for the analyses? The authors should elaborate on this point.

1.2 The authors should elaborate in the main text and also in the star methods on how they identified unannotated lncRNAs. What pipeline was used for their identification? Did the authors use coding potential algorithms to filter out potential peptide coding genes etc?

2. Identification of the molecular mechanism of action of lnc1845

2.1 The authors should mention if lnc1845 has been previously annotated or if it is a new lncRNA that is not present in Genocode or NONCODE databases (this is how the authors defined annotated lncRNAs in their study; see star methods). Given that the authors have RNA-seq data, it will be important to show RNA-seq genome browser tracks for lnc1845 right at the beginning when describing the locus. In general, one has to be particularly cautious with unspliced lncRNA transcripts such as lncRNA1845 as they often tend to be mis-mapping artefacts or pseudogenes. Showing RNA-seq genome tracks will also visualize the organization of the locus and demonstrate how are the levels of lnc1845 comparable to the expression levels of its protein-coding neighbor LHX8. Furthermore, the authors should include chromatin marks associated with the active enhancers around the lnc1845/LHX8 locus in the same figure. From Figure 6A it appears that the full deletion affects cis-regulatory elements – the authors should elaborate on this point.

2.2 The authors have presented RNA seq data analyses comparing a full deletion of the lncRNA locus to wild-type cells. In addition, to the full-deletion allele, the authors have introduced a premature transcriptional termination cassette which leads to efficient depletion of lnc1845. It will be important to compare RNA-seq data from full deletion, polyA knock-in, and wild-type alleles.

2.3 The authors mention that LHX8, the protein-coding gene regulated by lnc1845, has a crucial function in ovarian follicle development. Does the loss of lnc1845 and LHX8 have any consequences for the PGCLC cells i.e. any cellular phenotype/changes in cellular behavior? Or does the PGCLC system enable accessing the molecular signature only? The authors should mention this point.

2.4 A complex genome editing strategy was used to generate a null allele of lnc1845. The authors should provide a sufficient description of their editing strategy in the main text.

2.5 The experimental strategy for lnc1845 overexpression should be elaborated in the main text. Furthermore, the authors should demonstrate the correct subcellular localization of lnc1845 upon overexpression.

2.6 Because the authors identify WDR5 as lnc1845 interactor rather by educated guess than taking an unbiased approach, more controls are needed to demonstrate lnc1845-WDR5 interaction by RIP experiments (i.e. cytoplasmic lncRNAs). Also, what happens to the WDR5 protein in the lncRNA deletion/premature transcription termination alleles? Do the levels of WDR5 change? Does it act together with lnc1845 only at the local locus and if yes, how?

3. Identification of the FOXP3 transcription factor as a key regulator of gonad-specific lncRNAs including lnc1845.

3.1 The authors should elaborate in the main text on how the overexpression of FOXP3 was achieved.

3.2 The authors should indicate the location of the FOXP3 binding sites within the lnc1845 locus. Were these sites affected by any of the genome editing strategies used?

*Reviewer #2 (Recommendations for the authors):*

A better description of the dataset, including the developmental stages to which the cells correspond, used would help.

The description of methods for quantification and statistical analysis could be improved. For example (1) the settings used for Monocle should be provided, as setting properly data distribution with the expression family parameter of the newCellDataSet function is critical for computing robust pseudotimes. (2) how were ChIP-seq data normalized? Did the authors perform target/input log2ratio? (3) Methods for calculating expression correlation are missing. In addition, regarding data availability, information as to the bulk RNA-seq data generated in this study is lacking. A section "code availability" should also be included.

Please describe the parameters used for the classification of the biotypes in the mat and met, not only by citing the Luo et al. reference.

It is essential that the cellular model used to explore the function of LNC1845 is well characterized, and the identity of the differentiated germ cells (using the term ovarian follicle-like cells (FLCs) instead of late PGCLCs might avoid confusion) obtained from hESC clearly demonstrated. While FACs analysis demonstrates the expression of DAZL, the RNA-seq data could be further exploited through comparison (Unsupervised clustering, PCA) with other datasets (Jung et al.; gonadal cells, etc). Similarly, the transcriptomes of differentiated cells resulting from perturbed expression of LNC1845 should be exploited beyond differential gene expression analysis to explore the possibility of altered differentiation, due to abnormal LHX8 expression.

The authors should provide more information as to how they determined the 5' of the LNC1845 transcription unit, as this is critical for their functional analyses (deletion of the whole LNC1845 gene, CRISPR ON), and for search for transcription factors binding sites. Indeed 3', but not 5' RACE is performed.

Figure 4/S4: the RNA-FISH signal for LNC1845 is not convincing; I appreciate that it is not detected in control cells, but the signal is rather diffuse and appears enriched in the nucleolus. In addition, only a few nuclei are shown, which prevents robust conclusions to be drawn.

How is the genomic integrity of LHX8 validated in LNC KO, as the two genes are 5 kb apart?

While overall the data are in favor of a cis effect of LNC1845, I disagree that the CRISPR On and EIF1 promoter insertion strategies allow to disentangle cis from trans effect.

Line 312: OE of LHX8 not introduced.

---

## [Author Response]

Essential revisions:The two reviewers appreciated that the manuscript provides a very useful resource of human germ cell-specific lncRNAs for the community, and commended the authors for the impressive set of orthogonal approaches (KO, interrupted transcription, overexpression) they relied on to address the functional interactions between LNC1845 and its nearby gene LHX8. However, clarifications are needed to better support the claims, notably regarding the bioinformatic methodologies and the experimental strategies used for the functional genetic part.Three main points, outlined below, have been identified as requiring particular attention. Please address them in a revised version of your manuscript. In addition, answer the individual comments of the reviewers in a detailed rebuttal letter, and provide the text clarifications and additional figures they request to include in the manuscript.

We thank editors and reviewers for recognizing the value and significance of our study. We have addressed and answered the reviewer’s concerns and questions in the following paragraphs:

1. More insights into the cellular/phenotypic consequences of altered LHX8 expression is requested by the two reviewers, regarding the mention that LHX8 has a crucial function in ovarian development. One suggestion was to exploit the RNA-seq data to infer differentiation states of lnc1845 KO late PGCLCs. It seems also reasonable to ask the reviewers to provide information regarding the comparative efficiency of late PGCLC differentiation in presence or upon LHX8 downregulation (LNC1845 KO). The authors are welcome to provide other experimental means by which they could address the question of the functional consequences of the loss of lnc1845 and LHX8 on female PGCLC differentiation.

We appreciate the suggestions to further exploit the RNA-seq data to infer differentiation states of lnc1845 KO late PGCLCs. First, we have analyzed differentially expressed genes and GO terms in lnc1845 KO PGCLCS vs WT PGCLCS, and the results showed Figure 5D demonstrated that expression of many genes were also affected by LNC1845 deletion, especially many meiotic and oogenesis related genes are among them (Figure 5—figure supplement 1C). We also compared RNA-seq data of lnc1845 KO PGCLCs, WT late PGCLCs and in vivo hPGCs by Principal Component Analysis, and the results showed late PGCLCs differentiated from WT showed higher similarity to in vivo hPGCs than cells differentiated from lnc1845 KO cells (Figure 5F, Figure 5—figure supplement 1B).

2. More information should be given as to how the 5' end of LNC1845 and proximal regulatory regions were defined (TF binding sites? 5'RACE?). The experimental strategy for editing and for overexpression should also be better described. This information is critical to support the functional analyses by deletion and by CRISPR ON.

We analyzed the predicted LNC1845 sequences from NONCODE and GENCODE database (chr1:75,122,517-75,123,927), and our RNA-seq mapping track showed the mapping sequences covered chr1:75,122,766-75,123,909 (18bp less than the database). Combining the database sequences and our RNA-seq results, we marked the 75,123,927 at chromosome 1 as the 5' end of LNC1845, and we analyzed binding sequences of transcription factors in proximal regulatory regions in the 2000bp up-stream region of 5' end of LNC1845 and other meiotic specific lncRNAs.

Following editor and reviewer suggestions, we have added more detailed information regarding gene editing at LNC1845 in the main text and method sections. (Figure 4A, 5A, 5G, 6A)

3. The description of the datasets (developmental stages), methods for annotation of unknown lncRNAs, for quantification and for statistical analyses needs improvement.

We have improved the description of the datasets, methods for annotation of unknown lncRNAs, for quantification and for statistical analyses and added them into methods, respectively.

Reviewer #1 (Recommendations for the authors):My specific comments for each part of the study are summarized below:1. Identification of lncRNAs expressed in human gonads1.1 The first part of the manuscript annotating and cataloging lncRNAs in human germ cells requires some clarifications in respect of the number of identified lncRNAs. First, the authors should comment on the number of protein-coding genes in their data sets. Given that the human genome has in total 20.000-25.000 protein-coding genes, is it not surprising that about 14.000-16.000 protein-coding genes have been found to be expressed in gonads? Second, the same is valid for the number of lncRNAs that the authors identify to be expressed in these cells. Could it be that the high number of protein-coding and noncoding transcripts are due to a low expression threshold used for the analyses? The authors should elaborate on this point.

It may not be surprising to identify thousands of coding and noncoding genes in developing gonads given that the human genome has more than 20,000 coding genes. The main goal of showing the total number of expressed genes was to compare the expression level in germ cells versus somatic cells in the same developing gonads. The total number of coding and noncoding genes expressed in germ cells are more than that in somatic cells (Figure 1 A). Moreover, we compared these expressed genes in both male and female cells at each stage, so we delineated the expression levels of coding and noncoding genes in germ cell versus somatic cells of the same gonads throughout the developing stages (Figure 1C, Figure 1—figure supplement 1A,B). The lncRNAs and protein coding genes were considered as expressed gene if a transcript was detected more than 1 transcripts per million reads (TPM) and mapped in at least two samples (we defined this in the first paragraph of the result section). Thus, there may be relatively low and high expressed genes in the datasets. To cover as many expressed genes as possible in the overall analysis and to provide the most complete datasets for other researchers, we did not further delete the relatively low expressed genes.

1.2 The authors should elaborate in the main text and also in the star methods on how they identified unannotated lncRNAs. What pipeline was used for their identification? Did the authors use coding potential algorithms to filter out potential peptide coding genes etc?

We appreciate Reviewer #1 suggestion to elaborate on how we identified unannotated lncRNAs. The classification of annotated and unannotated lncRNA genes are performed by comparing the lncRNA genes’ genomic location between the NONCODEv5_hg38.lncAndGene.be (from NONCODE database) and gencode.v29.long_noncoding_RNAs.gtf (from Gencode database). A lncRNA is defined as annotated if it is included in these two databases, and as unannotated if it is only recorded in NONCODE database. GENCODE is the reference human genome annotation for The ENCODE Project and is widely used and contains more comprehensive annotation of lncRNA loci than UCSC and RefSeq. LncRNAs protein-coding potential was excluded through CNIT (Coding- NonCoding Identifying Tool), as all lncRNAs recorded in NONCODE database.

2. Identification of the molecular mechanism of action of lnc18452.1 The authors should mention if lnc1845 has been previously annotated or if it is a new lncRNA that is not present in Genocode or NONCODE databases (this is how the authors defined annotated lncRNAs in their study; see star methods). Given that the authors have RNA-seq data, it will be important to show RNA-seq genome browser tracks for lnc1845 right at the beginning when describing the locus. In general, one has to be particularly cautious with unspliced lncRNA transcripts such as lncRNA1845 as they often tend to be mis-mapping artefacts or pseudogenes. Showing RNA-seq genome tracks will also visualize the organization of the locus and demonstrate how are the levels of lnc1845 comparable to the expression levels of its protein-coding neighbor LHX8. Furthermore, the authors should include chromatin marks associated with the active enhancers around the lnc1845/LHX8 locus in the same figure. From Figure 6A it appears that the full deletion affects cis-regulatory elements – the authors should elaborate on this point.

Lnc1845 is an annotated lncRNA since we found the same location of it both in NONCODEv5 and GENCODEv29. Lnc1845 is known as NONHSAG001845.2 in NONCODE, and is named as AC099786.3 (ENSG00000261213.1) in GENCODE.

To show RNA-seq genome browser tracks for lnc1845 right at the beginning, we added the RNA- seq tracks to figure 5A with the deletion strategy.

To show the chromatin marks associated with the active enhancers around the lnc1845/LHX8 locus, we presented H3K4me3, H3K27ac ChIP-seq tracks along with RNA-seq tracks in Figure 6F. LNC1845 full deletion showed in figure 5A only removed the sequences of LNC1845, but not affected sequences of cis-regulatory elements of LHX8, which showed in the pink box in figure 6F.

2.2 The authors have presented RNA seq data analyses comparing a full deletion of the lncRNA locus to wild-type cells. In addition, to the full-deletion allele, the authors have introduced a premature transcriptional termination cassette which leads to efficient depletion of lnc1845. It will be important to compare RNA-seq data from full deletion, polyA knock-in, and wild-type alleles.

The main goal of the polyA knock-in is to validate that the expression of LHX8 is also downregulated using an independent method to inhibit the transcription of lnc1845, so we have not carried out the polyA knock-in RNA-seq due to limited funding for this project. We showed that the expression of LHX8 indeed significantly downregulated (Figure 5H), similar to the effect of full deletion in Figure 5B. Although we did not compare the RNA-seq of full deletion and polyA knockin to analyze the potential downstream effector genes and cellular pathway, we compared the RNA-seq of full deletion vs wild type, and LHX8 OE vs control. The comparison yielded many overlapped effector genes and GO term related to germ cell development and oogenesis (Figure 5—figure supplement 1B).

2.3 The authors mention that LHX8, the protein-coding gene regulated by lnc1845, has a crucial function in ovarian follicle development. Does the loss of lnc1845 and LHX8 have any consequences for the PGCLC cells i.e. any cellular phenotype/changes in cellular behavior? Or does the PGCLC system enable accessing the molecular signature only? The authors should mention this point.

We appreciate Reviewer #1 suggestion to further exploit the RNA-seq data to infer differentiation states of lnc1845 KO late PGCLCs. First, we have analyzed differentially expressed genes and GO terms in lnc1845 KO PGCLCS vs WT PGCLCS, and the results showed in Figure 5D demonstrated that expression of many genes were also affected by LNC1845 deletion, especially many germ cell development and oogenesis related genes are among them (Figure 5—figure supplement 1C). We also compared RNA-seq data of lnc1845 KO PGCLCs, WT late PGCLCs and in vivo hPGCs by Principal Component Analysis, and the results showed late PGCLCs differentiated from WT showed higher similarity to in vivo hPGCs than cells differentiated from lnc1845 KO cells (Figure 5F, Figure 5—figure supplement 1B).

2.4 A complex genome editing strategy was used to generate a null allele of lnc1845. The authors should provide a sufficient description of their editing strategy in the main text.

We appreciate Reviewer #1 suggestion to provide a sufficient description of the editing strategy, and we have added more detailed information for editing in main text and method section. In the method section under ‘CRISPR/Cas9 mediated knock-out and knock-in at LNC1845’, we have added ‘We analyzed the predicted LNC1845 sequences from NONCODE and GENCODE database (chr1:75,122,517-75,123,927), and our RNA-seq mapping track showed the mapping sequences covered chr1:75,122,766-75,123,909 (18bp less than the database).

Combining the database sequences and our RNA-seq results, we marked the 75,123,927 at chromosome 1 as the 5' end of LNC1845. Plasmids expressing Cas9/sgRNAs and donor sequences were cotransfected into H9 by lipofectamine 3000. The ratio of Cas9/sgRNAs versus donor is 1:4. For LNC1845 knockout, we designed a donor plasmid including two homology arms and two loxp locus flanking the sequence encoding puromycin drug resistance, replacing LNC1845 transcript but keeping the 5’ upstream and 3’ downstream of LNC1845 (Figure 5A).’ In the main text, we have added ‘First, we used a pair of gRNAs targeting near 5’ of LNC1845 locus and inserted drug selection sequences through homologous recombination. After drug selection for single colonies, we used Cre system to remove drug selection sequences.’

2.5 The experimental strategy for lnc1845 overexpression should be elaborated in the main text. Furthermore, the authors should demonstrate the correct subcellular localization of lnc1845 upon overexpression.

We have added more descriptions in addition to the illustration of strategy for lnc1845 overexpression in main text and main figures (Figure 6A, 6D). For cis overexpression, we used constitutive promoter insertion and co-expression of three different sgRNAs targeting the LNC1845 promoter region with a dCas9-VP64 activator. For trans overexpression, we used two different approaches, first by transfection of in vitro transcripts into PGCLCs and second by lentiviral transduction of PGCLCs. The LNC1845 sequences were inserted into the lentiviral vector p2k7 as reported in our previous study (Liang et al., 2019), and after the lentiviral packaging, the lentivirus was diluted from 5 folds to 100 folds for different concentration lentiviral infection.

Regarding the subcellular localization of LNC1845, we have added new experimental results using cellular fractionation assay and showed that the majority of LNC1845 was detected in nucleus, similar to other lncRNAs such as MALAT1 and NEAT1. In contrast, mRNA of GAPDH was mostly detected in cytoplasm of the same cells. (Related to Figure 6—figure supplement 1E, F.)

2.6 Because the authors identify WDR5 as lnc1845 interactor rather by educated guess than taking an unbiased approach, more controls are needed to demonstrate lnc1845-WDR5 interaction by RIP experiments (i.e. cytoplasmic lncRNAs). Also, what happens to the WDR5 protein in the lncRNA deletion/premature transcription termination alleles? Do the levels of WDR5 change? Does it act together with lnc1845 only at the local locus and if yes, how?

We have included GAPDH as the cytoplasmic mRNA and negative control because of the abundancy of the mRNA (Figure 6—figure supplement 2D). Most of the cis lncRNAs were not cytoplasmic and relatively low abundance compared to GAPDH, so the no detection of the cytoplasmic lncRNAs might not be feasible to be used as negative controls in the RIP experiments. In our RNA-seq data, the expression of WDR5 didn’t show significantly changes after lncRNA deletion (data not show). With our current datasets and the known functional roles of WDR5 in other studies with lncRNAs (Subhash et al., 2018; Wang et al., 2011; Yang et al., 2014), we proposed that LNC1845 directly interacts with WDR5 to modulate the H3K4me3 modification for LHX8 transcriptional activation. More experiments would be needed to validate all detailed mechanisms of this regulation.

3. Identification of the FOXP3 transcription factor as a key regulator of gonad-specific lncRNAs including lnc18453.1 The authors should elaborate in the main text on how the overexpression of FOXP3 was achieved.

We added more details in the main text as follow: First, we tested the effect of overexpressing FOXP3 in 293FT cells. The coding sequence of FOXP3 was inserted into a lentiviral vector with E1Fα promoter (more details in method) and transduced into 293FT cells. qPCR analysis showed that overexpression of FOXP3 alone upregulates LNC1845 and LHX8 expression, but TFAP2A alone did not show the upregulating effect (Figure 7—figure supplement 1E). Second, a modified overexpression vector was conducted in which FOXP3 was cloned and linked with P2A-mCherry in order to collect the overexpressed FOXP3 hPGCLCs by flow cytometry sorting for mCherry positive cells. FOXP3 overexpressed cells (FOXP3-P) showed higher expression of LNC1845 and LHX8 compared with the control FOXP3-N cells (Figure 7A).

3.2 The authors should indicate the location of the FOXP3 binding sites within the lnc1845 locus. Were these sites affected by any of the genome editing strategies used?

As shown in the alignment of tracks in Author response image 1, putative FOXP3 binding sites aligned with H3K4me3 ChIP and located at the genomic location 5’ upstream of LNC1845, so FOXP3 binding site is most likely located within the promoter (Author response image 1). This area was not deleted or modified by any of the genome editing strategies used in our studies and the FOXP3 sets of experiments (Figure 7, Figure 7—figure supplement 1) did not use any of the edited cell lines.

**Author response image 1. sa2fig1:** Integrative genomic viewer images showing FOXP3 binding sites at the LC1845/LHX8 locus.

Reviewer #2 (Recommendations for the authors):A better description of the dataset, including the developmental stages to which the cells correspond, used would help.The description of methods for quantification and statistical analysis could be improved. For example.1) The settings used for Monocle should be provided, as setting properly data distribution with the expression family parameter of the newCellDataSet function is critical for computing robust pseudotimes.

Single-cell pseudotime trajectories were constructed with the Monocle2 package (v2.10.1) according to the operation manual (http://cole-trapnell-lab.github.io/monocle-release/docs_mobile/). Briefly, the UMI count matrices of the germ cells were input as the expr_matrix, and meta_data was input as the sample sheet. Then, highly variable protein coding genes and lncRNAs identified by Seurat were chosen to define a cell’s progress. In this step, ordering genes that were expressed in less than 5 cells and had a P-value bigger than 0.001 were excluded. DDRTree was used to reduce the space down to one with two dimensions, and all cells were ordered with the orderCells function.

2) How were ChIP-seq data normalized? Did the authors perform target/input log2ratio?

We binned ChIP and Input into the same bins (10bp size), then normalized the reads overlapping the bins by RPKM for both ChIP and Input to account for differences in sequencing depth. After normalizing, we used the log2-ratio between the bins as a measurement of enrichment over input.

3) Methods for calculating expression correlation are missing. In addition, regarding data availability, information as to the bulk RNA-seq data generated in this study is lacking. A section "code availability" should also be included.Please describe the parameters used for the classification of the biotypes in the mat and met, not only by citing the Luo et al. reference.

We have described the data processing procedures in STAR METHODS:

Calculating lncRNA-mRNA pairs expression correlation were described in Analyzing single-cell RNA-seq Data part. The lncRNA-mRNA pairs expression correlation coefficient is calculated using R function cor() with method = "pearson". The cells used for expression correlation analysis are selected under the criteria that they must expressed the specific lncRNA of the lncRNA-mRNA pairs. RNA-Seq data processing were described in in vitro bulk RNA-seq library preparation, sequencing and data processing part. Raw sequencing reads were cleaned after removing low quality reads and trimmed adaptors. The quality of cleaned reads was assessed using FastQC and MultiQC. Next, clean reads were mapped to the Gencode.v29 of the human reference genome using Hisat2 with the GTF file contains both the annotations of protein coding genes from Gencode.v29 and lncRNA genes from NONCODEv5. Read counts per gene were obtained from the aligned reads using htseq-count of HTSeq package. R package DESeq2 was used for normalization and differentially expressed genes analysis.Code Availability was described as:

Custom scripts used in this study are publicly available at https://github.com/Maggie159123/*eLife*_2022.git.

It is essential that the cellular model used to explore the function of LNC1845 is well characterized, and the identity of the differentiated germ cells (using the term ovarian follicle-like cells (FLCs) instead of late PGCLCs might avoid confusion) obtained from hESC clearly demonstrated. While FACs analysis demonstrates the expression of DAZL, the RNA-seq data could be further exploited through comparison (Unsupervised clustering, PCA) with other datasets (Jung et al.; gonadal cells, etc). Similarly, the transcriptomes of differentiated cells resulting from perturbed expression of LNC1845 should be exploited beyond differential gene expression analysis to explore the possibility of altered differentiation, due to abnormal LHX8 expression.The authors should provide more information as to how they determined the 5' of the LNC1845 transcription unit, as this is critical for their functional analyses (deletion of the whole LNC1845 gene, CRISPR ON), and for search for transcription factors binding sites. Indeed 3', but not 5' RACE is performed.

We have analyzed differentially expressed genes and GO terms when lnc1845 KO, and the results showed in Figure 5B demonstrated that expression of many genes was also affected by LNC1845 deletion, and many meiotic and oogenesis, germ cell development related genes are among them (Figure 5—figure supplement 5C). We also compare RNA-seq data of lnc1845 KO PGCLCs, WT late PGCLCs and in vivo hPGCs by Principal Component Analysis, and the results showed late PGCLCs differentiated from WT showed higher similarity to in vivo hPGCs than cells differentiated from lnc1845 KO cells (Figure 5F, Figure 5—figure supplement 1B).

We analyzed the predicted LNC1845 sequences from NONCODE and GENCODE database (chr1:75,122,517-75,123,927), and our RNA-seq mapping track showed the mapping sequences covered chr1:75,122,766-75,123,909 (18bp less than the database). Combining the database sequences and our RNA-seq results, we marked the 75,123,927 at chromosome 1 as the 5' end of LNC1845, and we analyzed binding sequences of transcription factors in proximal regulatory regions in the 2000bp up-stream region of 5' end of LNC1845 and other meiotic specific lncRNAs.

Figure 4/S4: the RNA-FISH signal for LNC1845 is not convincing; I appreciate that it is not detected in control cells, but the signal is rather diffuse and appears enriched in the nucleolus. In addition, only a few nuclei are shown, which prevents robust conclusions to be drawn.How is the genomic integrity of LHX8 validated in LNC KO, as the two genes are 5 kb apart?

We have examined more RNA-FISH staining for LNC1845 and the nuclear localization and colocalization with LHX8 positive nuclei are consistent in multiple groups of PGCLCs in different area of the slide. We are not sure whether LNC1845 are enriched in nucleolus as other area of nuclei also have strong signal of LNC1845. In contrast, these signals are nondetectable or very weak in the control cells.

We have shown the mapping and DNA sequencing results at LNC1845 deleted regions in figure S6A. To further confirm the integrity of LHX8 sequence, we analyzed sequences between loxp and the first exon of LHX8 and found there was no deletion or other differences between WT and LNC1845 KO cells (data not show). The rest of LHX8 sequences were not altered because the RNA-seq data have detected the expression of LHX8 transcripts (Figure 5A).

While overall the data are in favor of a cis effect of LNC1845, I disagree that the CRISPR On and EIF1 promoter insertion strategies allow to disentangle cis from trans effect.

we agree with reviewer #2 that although the overall experimental evidences strongly support the model of cis effect, we are still not able to completely rule out the possibility of the trans effect.

Line 312: OE of LHX8 not introduced.

We have added introduction of LHX OE as follow: To further dissect LNC1845 function beyond regulating LHX8, we examined how similar are the gene expressions affected by LNC1845 and the gene expressions affected by overexpression of LHX8 in PGCLCs (LHX8 OE).

Reference:

1. Guttman M, Donaghey J, Carey BW, Garber M, Grenier JK, Munson G, Young G, Lucas AB, Ach R, Bruhn L, Yang X, Amit I, Meissner A, Regev A, Rinn JL, Root DE, Lander ES. lincRNAs act in the circuitry controlling pluripotency and differentiation. Nature. 2011 Aug 28;477(7364):295-300.

2. Luo S, Lu JY, Liu L, Yin Y, Chen C, Han X, Wu B, Xu R, Liu W, Yan P, Shao W, Lu Z, Li H, Na J, Tang F, Wang J, Zhang YE, Shen X. Divergent lncRNAs Regulate Gene Expression and Lineage Differentiation in Pluripotent Cells. Cell Stem Cell. 2016 May 5;18(5):637-52.

3. Subhash S, Mishra K, Akhade VS, Kanduri M, Mondal T, Kanduri C. H3K4me2 and WDR5 enriched chromatin interacting long non-coding RNAs maintain transcriptionally competent chromatin at divergent transcriptional units. Nucleic Acids Res. 2018 Oct 12;46(18):9384- 9400.

4. Li K, Xu J, Luo Y, Zou D, Han R, Zhong S, Zhao Q, Mang X, Li M, Si Y, Lu Y, Li P, Jin C, Wang Z, Wang F, Miao S, Wen B, Wang L, Ma Y, Yu J, Song W. Panoramic transcriptome analysis and functional screening of long noncoding RNAs in mouse spermatogenesis. Genome Res. 2021 Jan;31(1):13-26. doi: 10.1101/gr.264333.120. Epub 2020 Dec 16.